# WEAK-TO-STRONG DIFFUSION WITH REFLECTION

**Lichen Bai**[1]   **Masashi Sugiyama**[2,3]   **Zeke Xie**[1‡]

[1]xLeaF Lab, The Hong Kong University of Science and Technology (Guangzhou)
[2]RIKEN AIP    [3]The University of Tokyo
lbai292@connect.hkust-gz.edu.cn    zekexie@connect.hkust-gz.edu.cn

## ABSTRACT

The goal of generative diffusion models is to align the learned distribution with the real data distribution through gradient score matching. However, inherent limitations of current generative models lead to an inevitable gap between generated data and real data. To address this, we propose Weak-to-Strong Diffusion (**W2SD**), a novel framework that utilizes the estimated gap between existing weak and strong models (i.e., weak-to-strong gap) to bridge the gap between an ideal model and a strong model. By employing a reflective operation that alternates between denoising and inversion with weak-to-strong gap, W2SD steers latent variables along sampling trajectories toward regions of the real data distribution. W2SD is highly flexible and broadly applicable, enabling diverse improvements through the strategic selection of weak-to-strong model pairs (e.g., DreamShaper vs. SD1.5, good experts vs. bad experts in MoE). Extensive experiments demonstrate that W2SD significantly improves human preference, aesthetic quality, and prompt adherence, achieving significantly improved performance across various modalities (e.g., image, video), architectures (e.g., UNet-based, DiT-based, MoE), and benchmarks. For example, Juggernaut-XL with W2SD can improve with the HPSv2 winning rate up to **90%** over the original results. Moreover, the performance gains achieved by W2SD markedly outweigh its additional computational overhead, while the cumulative improvements from different weak-to-strong gap further solidify its practical utility and deployability.

## 1 INTRODUCTION

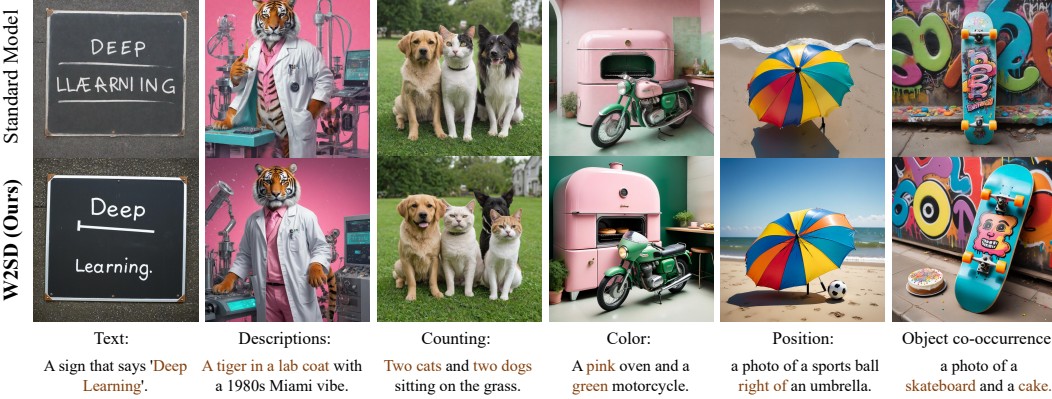

Text:

A sign that says 'Deep Learning'.

Descriptions:

A tiger in a lab coat with a 1980s Miami vibe.

Counting:

Two cats and two dogs sitting on the grass.

Color:

A pink oven and a green motorcycle.

Position:

a photo of a sports ball right of an umbrella.

Object co-occurrence:

a photo of a skateboard and a cake.

Figure 1: The qualitative results of standard Juggernaut-XL and its W2SD version, where Juggernaut-XL acts as a strong model and SDXL acts as a weak model. We present more cases in Appendix D.2.

In probabilistic modeling, the estimated density represents model-predicted distributions while the ground truth density reflects actual data distributions. Diffusion models (Song et al.), known for its powerful generative capabilities and diversity, estimate the gradient of the log probability densities in perturbed data distributions by optimizing a score-based network, which has become a mainstream paradigm in many generative tasks (Podell et al.; Guo et al., 2023; Shao et al., 2025a) and discriminative tasks (Li et al., 2025).

**Algorithm 1** W2SD

1: **Input:** Strong Model $\mathcal{M}^{\mathrm{s}}$, Weak Model $\mathcal{M}^{\mathrm{w}}$, Total Inference Steps: $T$, optimization steps: $\lambda$
2: **Output:** Clean Data $x_0$
3: Sample Gaussian noise $x_T$
4: **for** $t = T$ **to** 1 **do**
5:   **if** $t > T - \lambda$ **then**
6:     #W2SD with Reflection
7:     $\tilde{x}_t = \mathcal{M}^{\mathrm{w}}_{\mathrm{inv}}(\mathcal{M}^{\mathrm{s}}(x_t, t), t)$
8:   **end if**
9:   $x_{t-1} = M^{\mathrm{s}}(\tilde{x}_t, t)$
10: **end for**

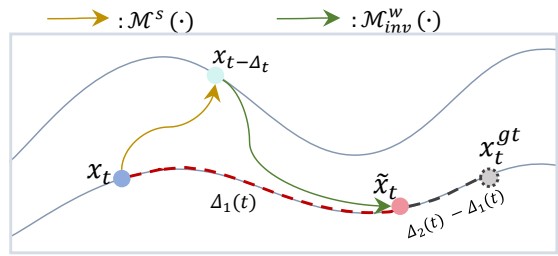

Figure 2: Visualizing the effectiveness of W2SD. When the weak-to-strong gap closely bridge the strong-to-ideal gap (e.g., $\Delta_2(t) - \Delta_1(t)$ is small), the refined latent variable $\tilde{x}_t$ converges to the ideal latent variable $x_t^{\mathrm{gt}}$.

Constrained by practical limitations including architectural design and dataset quality issues, existing diffusion models inevitably exhibit gradient estimation errors during inference stage, leading to a modeling gap between the real data distribution and the learned one. This gap is always a stubborn defect of diffusion models, often causing various problems (Sadat et al., 2024; Liu et al., 2024b; Aithal et al., 2024; Huang et al., 2025).

A large body of work has focused on advanced sampling techniques through input conditioning (Zhou et al., 2024; Ahn et al., 2024), architectural modifications (Si et al., 2024), and extra constraints (Chung et al., 2022; He et al., 2023; Sabour et al., 2024) to bridge this gap between an existing model and an ideal one that can perfectly recover real data. For example, Z-Sampling(Bai et al., 2024) improves sampling quality through implicit semantic injection, while Auto-Guidance (Karras et al., 2024) requires training a degraded version of the model from scratch to guide the sampling process. Despite these efforts, improvements are often confined to isolated components and focus on bridging the modeling gap from a single perspective (e.g., more ideal schedulers or denoising networks), leading to limited flexibility and generalization across different settings.

A natural question arises: can we flexibly leverage the various advantages of existing methods to further bridge the modeling gap and get closer to an ideal model(or a real data distribution)? Our answer is affirmative. In this work, we propose Weak-to-Strong Diffusion (W2SD), a meta-improving framework that flexibly integrates some existing diffusion enhancement technique (as the strong model) and its corresponding weak version into the sampling process, enabling training-free performance enhancement. To enable smoother and more effective gap bridging, departing from previous rough extrapolation methods (Karras et al., 2024), we draw upon a novel reflective operator into the diffusion sampling process in Figure 2, which is shown in Appendix E.5 to be both more flexible and powerful. Instead of improving specific components, W2SD can generally leverage various weak-to-strong gaps to achieve a stronger model with a bridged strong-to-ideal gap. We also intuitively illustrate the motivation in Figure 4. The contributions can be summarized as follows.

**First**, we introduce a general weak-to-strong concept into the inference enhancement of diffusion models by flexibly leveraging various weak-to-strong gaps to bridge the modeling gap between a strong model and the ideal mode, namely the real data distribution.

**Second**, we propose a novel W2SD framework with a reflection mechanism that alternately applies denoising with the strong model and inversion with the weak model[1], which theoretically enables our framework to implicitly estimate the weak-to-strong gap through iterative reflection, effectively steering latent variables along the sampling trajectory closer to the real data distribution.

**Third**, extensive experiments validate the effectiveness and broad applicability of W2SD across various generation tasks (e.g., image, video), architectures (e.g., UNet-based, DiT-based), and evaluation metrics. By extending the weak-to-strong concept, users can flexibly define weak-to-strong model pairs based on their specific needs. Depending on the type of model pair (e.g., DreamShaper vs. SD1.5; good experts vs. bad experts in MoE; ControlNet vs. DDIM), W2SD achieves diverse improvements in aspects such as human preference, prompt adherence, and personalization. Notably, these improvements are cumulative and complementary, further enhancing generation quality. We provide a detailed application analysis in Section 4. Efficiency evaluations also show that W2SD's

---

[1]In this work, we refer to such inversion as diffusion reflection, following Bai et al. (2024).

gains outweigh its computational overhead, maintaining superior quality under equal time constraints, which highlights its strong versatility and practical applicability.

## 2 PRELIMINARIES

In this section, we present preliminaries about denoising and inversion operation in diffusion models (Song et al.; Ho et al., 2020). Due to page limitations, we introduce more related works in Appendix B.

**Diffusion Model**  Given the random Gaussian noise $z_t$, we denote the forward process of diffusion models as $x_t = x_{t-\Delta t} + \sigma^t \sqrt{\Delta t} z_t$, where $t \in [0, 1]$, and $\sigma$ represents the predefined variance. We denote the ground truth density of real data distribution as $p_0^{\text{gt}}$, After noise addition at time $t$, the resulting density is represented as $p_t^{\text{gt}}$. Following DDIM (Song et al., 2020), we can obtain the denoised results $x_{t-\Delta t}$ from noisy data $x_t$ through the process of an ordinary differential equation as

$$x_{t-\Delta t} = \mathcal{M}(x_t, t) = x_t + \sigma^{2t} s_\theta(x_t, t) \Delta t, \quad t \in [0, 1], \tag{1}$$

where $s_\theta(\cdot, \cdot)$ represents the trained score network, utilized to predict the score at the time $t$. For simplicity, we set the noise schedule parameter $\sigma(t) = \sigma^t$. Similarly, here we can invert Equation (1) to transform $x_{t-\Delta t}$ back to a new $\tilde{x}_t$ as

$$\tilde{x}_t = \mathcal{M}_{\text{inv}}(x_{t-\Delta t}, t), \tag{2}$$

$$= x_{t-\Delta t} - \sigma^{2t} s_\theta(x_t, t) \Delta t, \tag{3}$$

$$\approx x_{t-\Delta t} - \sigma^{2t} s_\theta(x_{t-\Delta t}, t) \Delta t, \quad t \in [0, 1]. \tag{4}$$

Given that the approximation error in Equation (4) is negligible and the same score netowrk $s_\theta$, $\mathcal{M}$ and $\mathcal{M}_{\text{inv}}$ can be treated as mutually inverse mappings, thereby satisfying $\mathcal{M}_{\text{inv}}(\mathcal{M}(x_t, t), t) = x_t$. In Appendix E.3, we conduct a detailed analysis of the impact caused by this approximation error.

**Diffusion Sampling Enhancement**  As inference scaling laws (Wu et al., 2024) gain attention, increasing efforts (Bai et al., 2024; Guo et al., 2024; Ma et al., 2025) have been devoted to enhancing diffusion models at inference time. Some recent techniques such as Z-Sampling (Bai et al., 2024) and Auto-guidance (Karras et al., 2024) can be approximately interpreted as specific instances of our framework with less flexibility and improvement. Z-Sampling (Bai et al., 2024) implicitly injects semantic information by exploiting the gap between guidance scales, which can be viewed as an instance of W2SD, we provide more analysis in Appendix E.4. Auto-Guidance (Karras et al., 2024) relies on naive latent interpolation and training a degraded version from scratch. In contrast, we propose that a much broader range of strong-weak pairings can be leveraged in W2SD, not limited to slightly degraded models, but also including variations in input conditions and sampling strategies. Moreover, unlike Auto-Guidance that relies on CFG-like extrapolation and often introduces artifacts (Sadat et al., 2024), our smooth reflective mechanism effectively mitigates these issues (see Appendix E.5).

## 3 METHODOLOGY

In this section, we propose the Weak-to-Strong Diffusion framework and interpret its mechanism.

### 3.1 WEAK-TO-STRONG DIFFUSION

In this subsection, we integrate the weak-to-strong concept into diffusion model inference, introduce the W2S framework, and establish its theoretical understanding.

**The gap of Estimated Density Gradients**  In diffusion models, the goal is to minimize the gradients gap of log probability densities between estimated results and ground truth. However, due to the real data distribution is inaccessible, this gap cannot be directly quantified. To tackle this issue, we consider models of varying capacities: a strong model (with its corresponding denoising process denoted as $\mathcal{M}^{\text{s}}$ and the estimated density as $p^{\text{s}}$) and a weak model (similarly, $\mathcal{M}^{\text{w}}$ and $p^{\text{w}}$).

As shown in Figure 4, we define the weak-to-strong gap as $\Delta_1 = \nabla \log p^{\mathrm{s}} - \nabla \log p^{\mathrm{w}}$ and the strong-to-ideal gap as $\Delta_2 = \nabla \log p^{\mathrm{gt}} - \nabla \log p^{\mathrm{s}}$. By approximating $\Delta_2$ using the estimable $\Delta_1$, we indirectly reduce the gap between existing diffusion models and the ideal model, bringing the learned distribution closer to the real data distribution. In the next subsection, we introduce how to achieve this approximation from a reflective perspective.

**W2SD** Consider a strong model $\mathcal{M}^{\mathrm{s}}$ and a weak model $\mathcal{M}^{\mathrm{w}}$, we can optimize the sampling trajectories using the reflection operator $\mathcal{M}^{\mathrm{w}}_{\mathrm{inv}}(\mathcal{M}^{\mathrm{s}}(\cdot))$, as outlined in Algorithm 1. Through the iterative integration of strong model denoising and weak model inversion, we achieve a step-by-step reflective process during sampling process, refining the latent variable $x_t$ into an improved $\tilde{x}_t$.

Importantly, the choice of $\mathcal{M}^{\mathrm{s}}$ and $\mathcal{M}^{\mathrm{w}}$ significantly affects the direction of improvements effects. We summarize some promising weak-to-strong model pairs in Table 1 of Appendix C.3. And in Section 4 we present extensive application analyses and experiments, highlighting the powerful capabilities and flexibility of the proposed framework.

In Theorem 1, we demonstrate that W2SD refines latent variable $x_t$ toward the direction defined by the estimated weak-to-strong gap $\Delta_1(t)$. As shown in Figure 2, when the weak-to-strong gap closely approximates the strong-to-ideal gap (i.e., $\Delta_2(t) - \Delta_1(t)$ is small), the reflection mechanism of W2SD drives $x_t$ closer to the ideal $x_t^{\mathrm{gt}}$. The visualization analysis in Section 3.2 validates the correctness of our theory, and we provide a detailed proof in Appendix A.1. In Appendix E.2, we present comprehensive qualitative and quantitative evidence suggesting that $\Delta_1(t)$ effectively bridges $\Delta_2(t)$, providing empirical validation for Theorem 1, which also means that in Figure 2, the refined latent $\tilde{x}_t$ progressively approaches the ground truth latent $x_{gt}$.

**Theorem 1 (Theoretical Interpretation of W2SD)** *Suppose $x_t$ is the latent variable at time $t$, let $p_t^{\mathrm{s}}$ and $p_t^{\mathrm{w}}$ denote the probability density estimates derived from $\mathcal{M}^{\mathrm{s}}$ and $\mathcal{M}^{\mathrm{w}}$. The reflective operator $\mathcal{M}^{\mathrm{w}}_{\mathrm{inv}}(\mathcal{M}^{\mathrm{s}}(\cdot))$ refines $x_t$ to $\tilde{x}_t$ as*

$$\tilde{x}_t = x_t + \sigma^{2t}\Delta t(\nabla_{x_t} \log p_t^{\mathrm{s}}(x_t) - \nabla_{x_t} \log p_t^{\mathrm{w}}(x_t)), \tag{5}$$

*where $\Delta_1(t) = \nabla_{x_t} \log p_t^{\mathrm{s}}(x_t) - \nabla_{x_t} \log p_t^{\mathrm{w}}(x_t)$ means the w2s gap between $\mathcal{M}^{\mathrm{s}}$ and $\mathcal{M}^{\mathrm{w}}$ at time $t$.*

In Theorem 2, we also provide conditions for using $\Delta_1$ as a proxy for $\Delta_2$, with analytical validation. To further corroborate this, Figure 3 illustrates that given any two of the three models (weak, strong, or ideal), the third can be analytically derived to satisfy the theoretical constraints, providing visual verification of Theorem 2.

**Theorem 2 (Error Bound for W2SD Approximation)** *Let the ideal, strong, and weak models ($\phi \in \{\mathrm{gt}, s, w\}$) be infinite Gaussian Mixture Distributions $p^{\phi}(x) = \int w^{\phi}(\mu)\mathcal{N}(x; \mu, \Sigma)d\mu$. Assume the normalized weight deviations of $s$ and $w$ relative to $\mathrm{gt}$ are characterized by polynomial bias functions $B_s^k$ and $B_w^k$, respectively. For any $\epsilon > 0$, if the relative bias ratio satisfies $\left| B_w^k(\mu, x)/B_s^k(\mu, x) - 2 \right| \leq \epsilon$, then the error between the weak-to-strong gap $\Delta_1(x)$ and the strong-to-ideal gap $\Delta_2(x)$ is bounded by:*

$$|\Delta_1(x) - \Delta_2(x)| \leq C(x) \cdot \epsilon \cdot |\Delta_2(x)|, \tag{6}$$

*where $C(x) > 0$ is a deviation efficiency function depending on distribution properties (see Appendix A.2).*

Furthermore, referencing Theorem 3 (detailed in Appendix A.3), we extend our analysis from local gaps to global distribution alignment. We prove that under the relative bias condition of Theorem 2 (where error factor $\epsilon < 1$), W2SD achieves a strictly lower Fisher Divergence. The error reduction is quantified as $\mathcal{J}(p^{\mathrm{gt}}\|p^{\mathrm{w2sd}}) \leq \epsilon^2 \mathcal{J}(p^{\mathrm{gt}}\|p^{\mathrm{s}}) < \mathcal{J}(p^{\mathrm{gt}}\|p^{\mathrm{s}})$, confirming that the reflection operator systematically reduces the total score matching error.

Finally, we analyze the limitations of W2SD when the bias consistency condition is violated. Two primary failure modes are identified: (1) **Model Conflict** (Figure 22), where the weak and strong models exhibit opposing biases (i.e., bias ratio $< 0$), causing the W2S gap to contradict the true optimization direction; and (2) **Model Over-similarity** (Figure 23), where the weak model mimics the strong model too closely (i.e., bias ratio $\approx 1$ instead of 2). In this case, the resulting $\Delta_1$ is insufficient to bridge the gap to the ground truth.

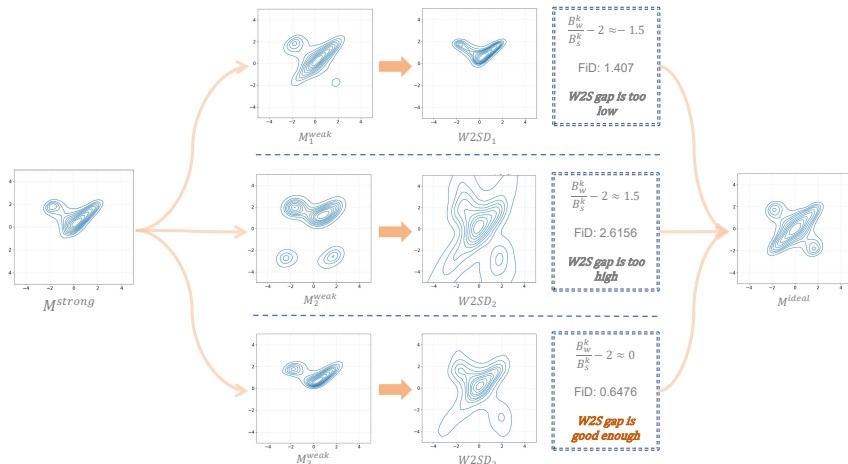

Figure 3: We modulate the weak model using Theorem 2, with the strong and ideal models fixed. A small $\epsilon$ allows the W2S gap to serve as a proxy for the S2I gap. For precise analysis, we derive the modulated weak model in closed form via diffusion model based on GMM data distribution.

## 3.2 VISUALIZATION AND INTERPRETATION

In this subsection, we validate the theory presented in Section 3.1 using both synthetic Gaussian mixture data and real-world image data, providing intuitive visual evidence.

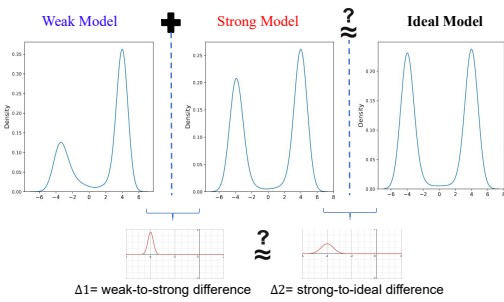

Figure 4: W2SD aims to bridge the strong model toward the ideal model by using $\Delta_1$ as a proxy for $\Delta_2$. In the 1-D case, we expect W2SD can shift the sampling distribution towards the left mode at "-4", increasing the likelihood of generating samples from low-density regions.

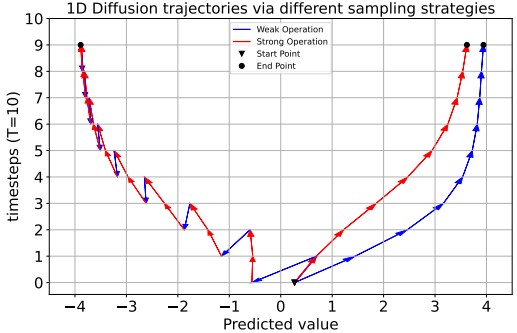

Figure 5: W2SD balances the distribution by integrating a weak model (blue) and a strong model (red) through a reflection mechanism, resulting in enhanced generation of samples in the left peak. We provide more cases in Figure 25.

**1-D Gaussian Mixture Data**  We begin by analyzing the 1-D Gaussian data scenario. In Figure 4, the diffusion model is designed to generate data with two distinct peaks at "-4" and "4". By adjusting the proportion of these two peaks in the training dataset, we obtain $\mathcal{M}^s$ and $\mathcal{M}^w$. In particular, $\mathcal{M}^s$ exhibits a clear bias toward the right peak, often failing to recover the left mode. In contrast, $\mathcal{M}^s$ demonstrates superior capability in capturing the structure of the left peak, indicating better generalization across modes. In Figure 5, we visualize the denoising trajectories under three different settings (weak model, strong model, W2SD). Through the reflective operator $\mathbf{M}_{\mathrm{inv}}^w(\mathcal{M}^s(\cdot))$, the latent variable is progressively drawn closer to the left peak. In contrast, for both strong and weak models, the generated samples are predominantly concentrated around the right peak.

**2-D Gaussian Mixture Data**  We also visualize the mechanism of W2SD on 2D scenario. In Figure 6 (first row), we modulate the proportion of the training data to obtain $\mathcal{M}^s$ and $\mathcal{M}^w$. W2SD balances the distribution by exploiting the discrepancy between $\mathcal{M}^s$ and $\mathcal{M}^w$ (the region in the bottom-left corner, denoted as $\Delta_1$) to bridge the unattainable strong-to-ideal gap $\Delta_2$, and boost

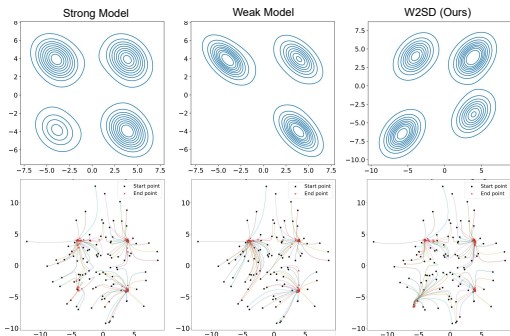

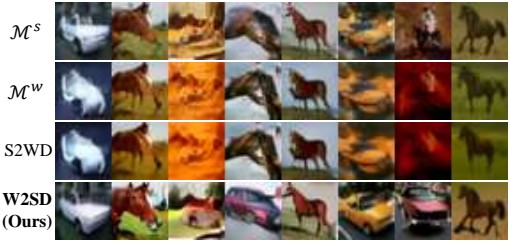

Figure 7: Qualitative results of W2SD based on dataset gap (CIFAR-10). Our framework enhances the probability of generating "cars" and promote a more balanced generation distribution.

Figure 6: Probability contour plot and denoising trajectories across different settings (2-D Gauss).

the chances of sampling toward the bottom-left region. In Figure 6 (second row), we visualize the denoising trajectories under different settings. W2SD balances the learned distribution, bringing it closer to the real data distribution. further validating the effectiveness of W2SD.

**Real Image Data** Furthermore, we investigate the performance of W2SD on CIFAR-10 (Krizhevsky et al., 2009). For ease of analysis, we select two classes from CIFAR-10 (Krizhevsky et al., 2009): **car** and **horse**. Specifically, the weak and strong diffusion models are trained on distinct datasets. Both $\mathcal{M}^s$ and $\mathcal{M}^w$ are biased toward generating horses. Moreover, since $\mathcal{M}^w$ is trained on only a limited number of car images, it rarely generates cars. In Figure 7, W2SD can help balance the image categories, enhancing inference process to increase the probability of generating cars. In Figure 18, we also perform a t-SNE dimensionality reduction (Van der Maaten & Hinton, 2008) on the CLIP features of the generated images. W2SD effectively disentangles the representations of "car" and "horse" in the 2D space. We proivde more details in Appendix E.2

## 4 EMPIRICAL ANALYSIS

In this section, we justify our design choices in Section 3 and illustrate the wide applicability and impressive improvement of W2SD across diverse weak-to-strong combinations of $\mathcal{M}^s$ and $\mathcal{M}^w$. Due to page limitations, we present more quantitative and qualitative results across diverse benchmarks (e.g., Drawbench (Saharia et al., 2022)) and modalities (e.g., video generation) in Appendix D.

**Experimental Setting** We present experimental results across three diverse model gap settings. For clarity, a summary of these meta-experimental results is provided in Table 1. For reliable evaluation, we use configuration-specific metrics: HPS v2 and PickScore for human preference alignment; AES for aesthetic quality; CLIP-T and CLIP-I for personalized generation in LoRA models; and FID & Inception Score (IS) for MoE expert selection strategies, measured against the training data distribution. We present more experimental details in Appendix C.

Table 1: Different types of model gap lead to improvement effects in different directions.

| Model Gap | $\mathcal{M}^s$ | $\mathcal{M}^w$ | Results |
|---|---|---|---|
| Weight Gap | Finetune Mechanism | SDXL/SD1.5 | Tables 2 and 8 |
| | LoRA Mechanism | SDXL/SD1.5 | Tables 3, 10 and 11 |
| | Strong Experts (MoE) | Weak experts (MoE) | Table 4 |
| Condition Gap | High CFG | Low CFG | Tables 5 and 9 |
| | Refined Prompt | Raw Prompt | Table 6 |
| Sampling Pipeline Gap | ControlNet | Standard Pipeline (DDIM) | Figure 11 |
| | IP-Adapter | Standard Pipeline (DDIM) | Figure 10 |

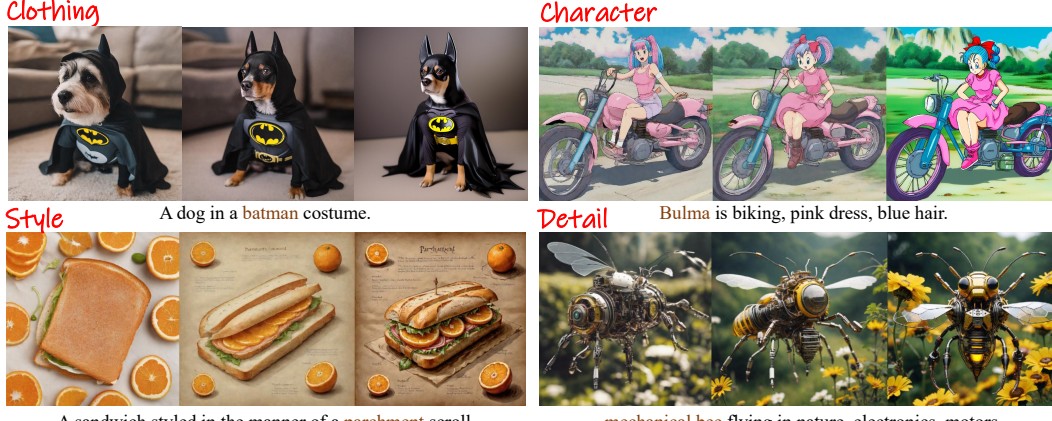

Figure 8: Qualitative comparisons with weak model (**left**), strong model (**middle**) and W2SD based on weight gap (**right**). Our framework utilizes the gaps between chosen strong and weak models (e.g., high-detail LoRA vs. standard model) to deliver improvements in various dimensions, including style, character, clothing, and beyond. We provide more qualitative results in Appendix D.2.

## 4.1 WEIGHT GAP

The capability gap between models can be directly captured by their parameter weights. And W2SD leverages this weight gap to empirically bridge the strong-to-ideal gap, enabling effective reflective operations.

Table 2: Quantitative results of W2SD based on a full parameter fine-tuning strategy. Our framework generates results better aligned with human preferences. Datasets: Pick-a-Pic.

| Method | HPS v2 ↑ | AES ↑ | PickScore ↑ | MPS ↑ |
|---|---|---|---|---|
| SD1.5 | 24.9558 | 5.5003 | 20.1368 | - |
| DreamShaper | 30.1477 | 6.1155 | 21.5035 | 46.8705 |
| **W2SD** | **30.4924** | **6.2478** | **21.5727** | **53.1304** |
| SDXL | 29.8701 | **6.0939** | 21.6487 | - |
| Juggernaut-XL | 31.6412 | 5.9790 | 22.1903 | 45.7397 |
| **W2SD** | **32.0992** | 6.0712 | **22.2434** | **54.2634** |

Table 3: Quantitative results of W2SD based on personalized LoRA model. Here, We set SD1.5 as weak model $\mathcal{M}^w$ and (SD1.5 + LoRA) as strong model $\mathcal{M}^s$. We note different LoRA modules enable distinct personalized outcomes, as shown in Figure 8 and Appendix D.2.

| Method | DINO ↑ | CLIP-I ↑ | CLIP-T ↑ |
|---|---|---|---|
| SD1.5 | 27.47 | 52.08 | 20.14 |
| Personalized LoRA | 48.03 | 64.37 | 25.99 |
| **W2SD** | **51.58** | **68.04** | **27.66** |

**Full Parameter Fine-tuning**   We first select the full parameter fine-tuned models (e.g., DreamShaper, Juggernaut-XL) that align more closely with human preferences as $\mathcal{M}^s$, and the corresponding standard models as $\mathcal{M}^w$ (e.g., SD1.5 (Rombach et al., 2022), SDXL (Podell et al.)). We evaluate our framework with various metrics, as shown in Table 2, W2SD based on weight gap shows significant improvements in human preference metrics such as HPS v2 (Wu et al., 2023b) and PickScore (Kirstain et al., 2023). We provide more visual cases in Figure 15 of Appendix D.2.

**LoRA Mechanism**   Furthermore, W2SD is also applicable to efficiently fine-tuned model. we select the personalized models derived through the LoRA mechanism (Hu et al.) as $\mathcal{M}^s$, and employ W2SD to attain a more robust and customized personalization effect. We test 20 LoRA checkpoints across object, person, animal, and style categories, with details in Appendix C.3. Table 3 demonstrates that W2SD results in significant improvements across multiple personalization metrics (e.g., Clip-T and Clip-I). We present the qualitative results in Figure 8, with more visual cases in Appendix D.2.

**MoE Mechanism**   We also note that a weak-to-strong gap can be induced by controlling the expert selection strategy within the MoE mechanism. Specifically, we focus on DiT-MoE (Fei et al., 2024), a novel architecture that integrates multiple experts and selects the two highest-performing experts

during each denoising step to optimize generation quality. Based on this framework, we use DiT-MoE-S as $\mathcal{M}^s$ and define $\mathcal{M}^w$ as the two lowest-performing experts in each denoising step, thereby establishing a quantifiable weight gap.

In Table 4, we show that W2SD reduces FiD (Seitzer, 2020) from 15.1032 to **9.1001**, aligning the learned distribution closer to the real data distribution. Additionally, although the limited capacity of MoE-DiT-S (with only 71M active parameters) often results in image degradation (the first row in Figure 9), the application of W2SD significantly enhances the image quality (the second row in Figure 9).

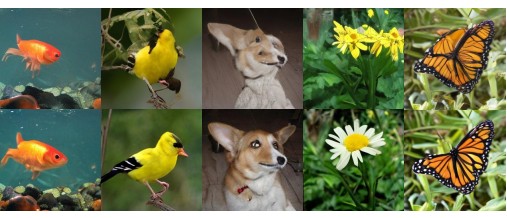

Figure 9: Quantitative Results of W2SD Based on the MoE Mechanism. The **first row** shows the results for DiT-MoE-S, while the **second row** presents the results of W2SD. We note W2SD achieves significant improvements, even with small models featuring 71M activated parameters, effectively eliminating image distortion.

Table 4: Quantitative Results of W2SD Based on MoE Mechanism. Dataset: ImageNet 50K.

| Method | IS ↑ | FiD ↓ | AES ↑ | HPS v2 ↑ |
|---|---|---|---|---|
| DiT-MoE-S | 45.4437 | 15.1032 | 4.4755 | 20.0486 |
| **W2SD** | **55.5341** | **9.1001** | **4.5053** | **22.3225** |

Table 5: Quantitative Results of W2SD Based on Guidance Gap. Dataset: Pick-a-Pic.

| Method | HPS v2 ↑ | AES ↑ | PickScore ↑ | MPS ↑ |
|---|---|---|---|---|
| SD1.5 | 24.9558 | 5.5003 | 20.1368 | 42.1101 |
| **W2SD** | **25.5069** | **5.5073** | **20.2443** | **57.8903** |
| SDXL | 29.8701 | 6.0939 | 21.6487 | 43.9425 |
| **W2SD** | **31.2020** | **6.0970** | **21.7980** | **56.0608** |

## 4.2 CONDITION GAP

Here we demonstrate that the weak-to-strong gap applies not only across different weights but also within the same diffusion model under the weak-to-strong conditions.

**Guidance Scale** The classifier-free guidance mechanism (Ho & Salimans, 2021) extrapolates between conditional and unconditional noise predictions, adjusting the guidance scale to control the semantic attributes of the generated images. To apply W2SD, diffusion models under high guidance scale can be viewed as $\mathcal{M}^s$, those under low or even negative guidance scale can be treated as $\mathcal{M}^w$. This guidance scale discrepancy constructs a weak-to-strong gap that aligns the learned generated distribution more closely with the semantic conditions of the prompt. In Table 5, our framework significantly enhances human preference (e.g., HPS v2) and aesthetic characteristics (e.g., AES) in mainstream models such as SDXL and SD1.5.

Table 6: Quantitative results of W2SD based on semantic gaps between prompts. Model: SDXL. Datasets: GenEval.

| Prompt Type | HPS v2 ↑ | AES ↑ | PickScore ↑ | MPS ↑ |
|---|---|---|---|---|
| Raw Prompt | 25.3897 | 5.4454 | 20.7144 | - |
| Refined Prompt | 28.5698 | 5.7714 | 21.6350 | 45.7719 |
| **W2SD** | **29.4023** | **5.8812** | **21.8053** | **54.2275** |

Table 7: The improvements effects from different model gaps can be cumulative. Datasets: Pick-a-Pic.

| Weight Gap | Condition Gap | HPS v2 ↑ | Winning Rate ↑ |
|---|---|---|---|
| × | × | 31.6412 | - |
| × | ✓ | 32.8217 | 84% |
| ✓ | × | 32.0992 | 76% |
| ✓ | ✓ | **32.9623** | **90%** |

**Semantics of Prompts** Aside from adjusting the guidance scale, we demonstrate that the semantic gaps within the condition prompts themselves can also be leveraged to enhance generation quality through the reflection process. Specifically, we select a total of 533 text prompts from the GenEval (Ghosh et al., 2024) for $\mathcal{M}^w$, with an average length ranging from 5 to 6 words and minimal contextual complexity. Additionally, we leverage LLM (here we use Qwen-Plus (Yang et al., 2024)) to enrich and refine these prompts with additional details for $\mathcal{M}_s$. resulting in weak-to-strong prompt pairs. We report the quantitative results in Table 6, which relies solely on the semantic gaps between prompts, achieves improvements across all metrics. And in Figure 16 of Appendix D.2 we show that W2SD is capable of more precisely capturing the details based on the gaps in prompt pairs.

### 4.3 SAMPLING PIPELINE GAP

Extensive works (Si et al., 2024; Chefer et al., 2023; Zhang et al., 2023) have been devoted to design advanced inference pipelines to improve the quality of diffusion generation results. We demonstrate that W2SD can enhance the inference process by leveraging the gap derived from these powerful existing pipelines. Here we define those enhanced sampling methods as $\mathcal{M}^s$, while $\mathcal{M}^w$ represents the standard sampling (Song et al., 2020; Lu et al., 2022).

We first select ControlNet (Zhang et al., 2023) as $\mathcal{M}^s$, which incorporates additional network structures into the pipeline. By utilizing reference images (e.g., edge maps), it facilitates controllable image generation. As shown in Figure 11, the images generated by W2Sd exhibit a stronger alignment with the provided edge maps. We also present visualization results of strong models such as IP-Adapter(Ye et al., 2023), as shown in Figure 10, with DDIM serving as the weak model. It can be observed that W2SD better aligns the generated results with the reference image. Therefore, our framework is largely orthogonal to efforts in the research community aimed at improving diffusion sampling, which significantly enhances the broad applicability of W2SD and further generalizes the weak-to-strong concept beyond (Karras et al., 2024).

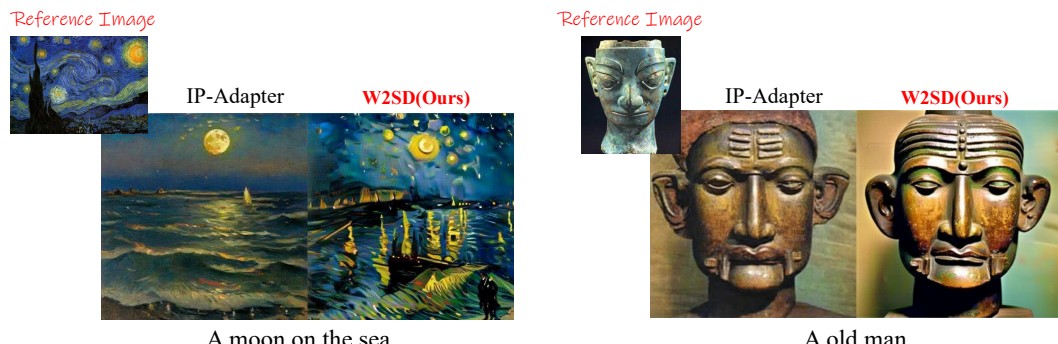

Figure 10: Qualative Results of W2SD based on pipeline gap. We set IP-Adapter as $\mathcal{M}^s$, DDIM as $\mathcal{M}^w$. W2SD makes the output better aligned with the reference image

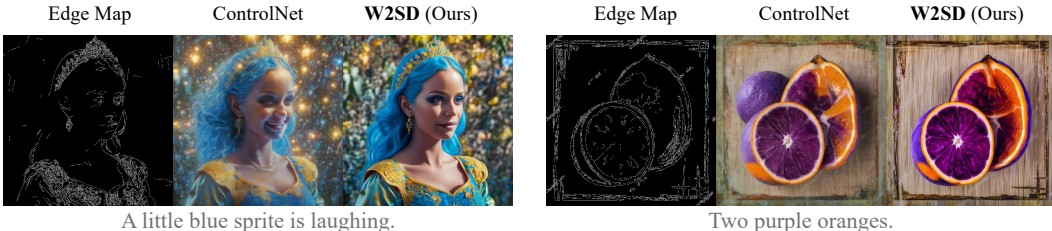

Figure 11: Results of W2SD based on pipeline gap. We set ControlNet as $\mathcal{M}^s$, DDIM as $\mathcal{M}^w$. W2SD makes the output better aligned with the edge map.

### 4.4 CUMULATIVE EFFECTS OF DIFFERENT MODEL GAPS

Finally, we note that the improvements effects of W2SD, derived from different type of gaps, can be even cumulative. Here we apply the weight gap and the condition gap simultaneously, where $\mathcal{M}^s$ represents the fine-tuned model Juggernaut-XL with high guidance scale at 5.5, while $\mathcal{M}^w$ represents the standard model SDXL with low guidance scale at zero. As shown in Table 7, the combination of weight gap and condition gap leads to a substantial improvement in Pick-a-Pic compared to $\mathcal{M}^s$, achieving a winning rate of up to 90% on HPS v2.

## 5 DISCUSSION AND ANALYSIS

In this section, we discuss and perform more detailed studies on W2SD.

**The Magnitude of Weak-to-Strong Gap**  In this subsection, we explore how the magnitude of weak-to-strong gap affects the improvements effects. To quantify the gap between $\mathcal{M}^{\text{s}}$ and $\mathcal{M}^{\text{w}}$, we first consider the LoRA-based W2SD framework. We fix the LoRA scale of $\mathcal{M}^{\text{s}}$ at 0.8 and adjust the LoRA scale of $\mathcal{M}^{\text{w}}$ (e.g., from -3.5 to 3.5) to observe its impact on the effects of improvements. As shown in Figure 12 (left), when the LoRA scale of $\mathcal{M}^{\text{s}}$ exceeds that of $\mathcal{M}^{\text{w}}$, W2SD improves performance. However, when $\mathcal{M}^{\text{s}}$ is weaker than $\mathcal{M}^{\text{w}}$ (i.e., the scale gap is negative), the improvements effects result in the negative gains.

In addition, as shown in Figure 12 (right), we also quantify the weak-to-strong gap based on guidance scale. By fixing the guidance scale of $\mathcal{M}^{\text{s}}$ at 5.5 and adjusting that of $\mathcal{M}^{\text{w}}$ (e.g., from -10 to 15), we observe similar phenomena. In Figure 13, we present a qualitative analysis showing that the magnitude of the weak-to-strong gap is a key factor to influence the generation quality.

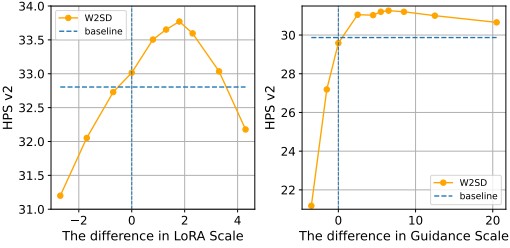
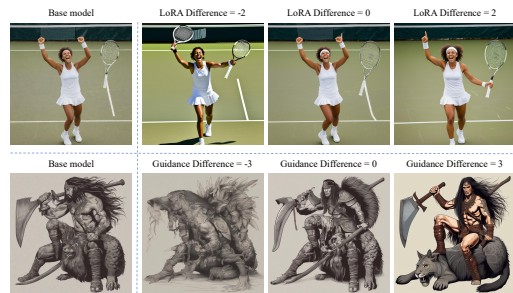

Figure 12: The magnitude of weak-to-strong gap is a key factor impacting the effects of improvements. The horizontal axis shows the magnitude of the weak-to-strong gap, while the vertical axis shows the average HPS v2 on the Pick-a-Pic. When $\mathcal{M}^{\text{s}}$ is weaker than $\mathcal{M}^{\text{w}}$, W2SD results in negative gains.

Figure 13: When the weak-to-strong gap is greater than 0, W2SD yields positive gains. When it equals 0, the process degenerates into standard sampling. When it is less than 0, negative gains occurs, resulting in poor image quality.

**Time Efficiency Comparison**  Assuming standard sampling and W2SD require $T_{\text{std}}$ and $T_{\text{w2s}}$ denoising steps, respectively, the score predictions needed are $T_{\text{std}}$ and $T_{\text{w2s}} + 2\lambda$. To ensure a fair sampling time efficiency comparison, we set $T_{\text{w2s}} = \lfloor \frac{1}{2} T_{\text{std}} \rfloor$ and $\lambda = \lfloor \frac{1}{2} T_{\text{w2s}} \rfloor$, matching the runtime for generating the same image. For example, if $T_{\text{std}} = 50$, standard sampling performs 50 denoising steps (e.g., DDIM), while W2SD uses $T_{\text{w2s}} + 2\lambda = 24 + 2 \cdot 12 = 48$, indicating that the time required to generate a single image is approximately equal. In this setting, as shown in Figure 14, W2SD consistently outperforms standard sampling, demonstrating that the gains from reflection operations far outweigh the additional computational cost, validating the time efficiency of our framework.

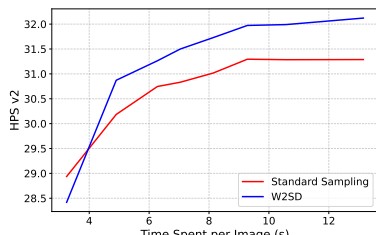

Figure 14: W2SD can outperform standard sampling by a large margin even under the same inference time with various inference steps.

For a more comparable analysis, we also examine the performance of W2SD under different settings of $T$ and $\lambda$, as shown in 19 and 20. The results demonstrate that performance consistently improves with increasing $\lambda$ for W2SD, a time inference scaling method, across both turbo and classical models.

## 6  CONCLUSION

In this work, we systematically integrate the weak-to-strong mechanism into the inference enhancement of diffusion models. We theoretically and empirically indicate that the estimated weak-to-strong gap can effectively bridge the strong-to-ideal gap, enhancing the alignment between the learned distributions from existing diffusion models and the real data distribution. Building on this concept, we propose W2SD, which utilizes the estimated gap in density gradients to optimize sampling trajectories via reflective operations. W2SD demonstrates its effectiveness as a general-purpose framework through its cumulative performance improvements, flexible definition of weak-to-strong model pairs, and significant performance gains with minimal computational overhead. We believe that our work that leverages weak-to-strong model pairs will motivate interesting works for more tasks in future.

ETHICS STATEMENT

This work introduces W2SD, a novel framework designed to enhance the inference process of diffusion models. The data and models utilized in our work are released under open-source licenses and sourced from open platforms. While our work may have various societal implications, it does not introduce additional ethnic concerns compared to existing standard sampling methods. As such, none which we feel must be specifically highlighted here.

ACKNOWLEDGEMENT

This work was supported by the National Natural Science Foundation of China under Grant No. 62506317 and Dream Set Off - Kunpeng and Ascend Seed Program. MS was supported by JST ASPIRE Grant Number JPMJAP2405.

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

# A PROOFS

In this section, we provide the proofs for Theorem 1 presented in this work.

## A.1 PROOF OF THEOREM 1

We first establish the relationship between the latent variable $x_t$ and the refined latent variable $\tilde{x}_t$ through the reflection operation, proving that the core mechanism of W2SD is to optimize $x_t$ in the direction of the weak-to-strong gap.

**Proof 1** *At a given time $t$, the W2SD reflection operation applies the operator $\mathcal{M}_{inv}^w \mathcal{M}^s(\cdot)$ to the latent variable $x_t$.*

*Specifically, we first denoise $x_t$ using the strong model $\mathcal{M}^s$ according to Equation (1), obtaining $x_{t-\Delta t}$ as*

$$x_{t-\Delta t} = x_t + \sigma^{2t} s_\theta^s(x_t, t)\Delta t, \tag{7}$$

*where $s_\theta^s$ denotes the score predicted by $\mathcal{M}^s$, which is equivalent to the gradient of the log probability density $\log p_t^s$, so, Equation (7) can be rewritten as*

$$x_{t-\Delta t} = x_t + \sigma^{2t}\nabla_{x_t} \log p_t^s(x_t)\Delta t. \tag{8}$$

*After obtaining $x_{t-\Delta t}$, we apply the weak model $\mathcal{M}^w$ to invert $x_{t-\Delta t}$ back to the noise level at time $t$ according Equation (4), thereby completing the reflection process as*

$$\tilde{x}_t \approx x_{t-\Delta t} - \sigma^{2t} s_\theta^w(x_{t-\Delta t}, t)\Delta t, \tag{9}$$

*where $s_\theta^w$ denotes the score predicted by $\mathcal{M}^w$. Since $\Delta t$ is typically small, we neglect the approximation error in the diffusion inversion process which implies that $s_\theta^w(x_{t-\Delta t}, t)$ is equivalent to the gradient of the log probability density $\log p_t^w$. Hence we can reformulate Equation (9) as*

$$\tilde{x}_t = x_{t-\Delta t} - \sigma^{2t}\nabla_{x_t} \log p_t^w(x_t)\Delta t. \tag{10}$$

*Combining Equation (8) and Equation (10), we obtain $\tilde{x}_t$ as*

$$\tilde{x}_t = x_{t-\Delta t} - \sigma^{2t}\nabla_{x_t} \log p_t^w(x_t)\Delta t \tag{11}$$

$$= x_t + \sigma^{2t}\nabla_{x_t} \log p_t^s(x_t)\Delta t - \sigma^{2t}\nabla_{x_t} \log p_t^w(x_t)\Delta t \tag{12}$$

$$= x_t + \sigma^{2t}\Delta t(\underbrace{\nabla_{x_t} \log p_t^s(x_t) - \nabla_{x_t} \log p_t^w(x_t)}_{\text{weak-to-strong gap}\Delta_1(t)}). \tag{13}$$

*From Equation (13), we observe that for the latent variable $x_t$, the reflection operator $\mathcal{M}_{inv}^w(\mathcal{M}^s(\cdot))$ in W2SD perturbs $x_t$ along the direction of $\Delta_1(t)$, producing the refined variable $\tilde{x}_t$. When the weak-to-strong gap $\Delta_1$ closely bridges the unattainable strong-to-ideal gap $\Delta_2(t)$, the resulting $\tilde{x}_t$ becomes more aligned with the ground truth ideal distribution.*

## A.2 PROOF OF THEOREM 2

We now establish the theoretical connection between the weak-to-strong gap $\Delta_1$ and the strong-to-ideal gap $\Delta_2$, providing the strict condition for W2SD to work effectively. It is worth noting that for $\Delta_1(t, x)$ and $\Delta_2(t, x)$ at time step $t$ with input $x$, we simplify the notation to $\Delta_1(x)$ and $\Delta_2(x)$ for convenience, with the understanding that the analysis is conducted at step $t$ by default.

**Proof 2** *Consider three distributions represented by infinite Gaussian mixture models:*

$$p^{gt}(x) = \int w^{gt}(\mu)\mathcal{N}(x; \mu, \Sigma)d\mu,$$

$$p^s(x) = \int w^s(\mu)\mathcal{N}(x; \mu, \Sigma)d\mu, \tag{14}$$

$$p^w(x) = \int w^w(\mu)\mathcal{N}(x; \mu, \Sigma)d\mu,$$

*where all models share the same covariance matrix $\Sigma$, representing a common level of distribution concentration. Since infinite GMMs can theoretically approximate any continuous distribution, this assumption is well-founded and provides universal approximation capability for our theoretical framework.*

*Define the posterior weight function:*

$$\gamma(\mu|x) = \frac{w(\mu)\mathcal{N}(x; \mu, \Sigma)}{p(x)}, \tag{15}$$

*and Equation (15) represents the probability that a given data point $x$ was generated from the Gaussian component centered at $\mu$, conditioned on the current observation.*

*The score function can then be expressed as*

$$\nabla_x \log p(x) = -\Sigma^{-1} \left[ x - \int \mu \gamma(\mu|x) d\mu \right], \tag{16}$$

*which reveals an intuitive geometric interpretation: the score function points from the current position $x$ toward the posterior mean $\mathbb{E}[\mu|x]$, with the covariance matrix $\Sigma^{-1}$ controlling the strength and direction of this attraction.*

*The ideal gap and computable gap are respectively as*

$$\begin{aligned}
\Delta_2(x) &= -\Sigma^{-1} \int \mu \left[ \gamma^{gt}(\mu|x) - \gamma^s(\mu|x) \right] d\mu \\
&= -\Sigma^{-1} \int \mu \left[ (\frac{w^{gt}(\mu)}{p^{gt}(x)} - \frac{w^s(\mu)}{p^s(x)}) \mathcal{N}(x; \mu, \Sigma) \right] d\mu,
\end{aligned} \tag{17}$$

$$\begin{aligned}
\Delta_1(x) &= -\Sigma^{-1} \int \mu \left[ \gamma^s(\mu|x) - \gamma^w(\mu|x) \right] d\mu \\
&= -\Sigma^{-1} \int \mu \left[ (\frac{w^s(\mu)}{p^s(x)} - \frac{w^w(\mu)}{p^w(x)}) \mathcal{N}(x; \mu, \Sigma) \right] d\mu,
\end{aligned} \tag{18}$$

*Specifically, to facilitate the analysis, we define*

$$\begin{aligned}
\frac{w^s(\mu)}{p^s(x)} &= \frac{w^{gt}(\mu)}{p^{gt}(x)} \cdot \left( 1 + \sum_{k=0}^{K} B_s^k(\mu, x) \right), \\
\frac{w^w(\mu)}{p^w(x)} &= \frac{w^{gt}(\mu)}{p^{gt}(x)} \cdot \left( 1 + \sum_{k=0}^{K} B_w^k(\mu, x) \right),
\end{aligned} \tag{19}$$

*where $B_s^k$ and $B_w^k$ represent the $k$-th order polynomial coefficients for the strong and weak models relative to the ideal model, respectively, and $K$ is the total polynomial order.*

*Building upon Equation (17) and Equation (19), we reformulate $\Delta_2(x)$ as:*

$$\begin{aligned}
\Delta_2(x) &= -\Sigma^{-1} \int \mu \left[ \frac{w^{gt}(\mu)}{p^{gt}(x)} - \frac{w^s(\mu)}{p^s(x)} \right] \mathcal{N}(x; \mu, \Sigma) d\mu \\
&= -\Sigma^{-1} \int \mu \left[ \frac{w^{gt}(\mu)}{p^{gt}(x)} - \frac{w^{gt}(\mu)}{p^{gt}(x)} \cdot \left( 1 + \sum_{k=1}^{K} B_s^k(\mu, x) \right) \right] \mathcal{N}(x; \mu, \Sigma) d\mu \\
&= \frac{\Sigma^{-1}}{p^{gt}(x)} \int \mu w^{gt}(\mu) \left( \sum_{k=1}^{K} B_s^k(\mu, x) \right) \mathcal{N}(x; \mu, \Sigma) d\mu
\end{aligned} \tag{20}$$

*Similarly, $\Delta_1(x)$ can be expressed as:*

$$\Delta_1(x) = -\Sigma^{-1} \int \mu \left[ \frac{w^s(\mu)}{p^s(x)} - \frac{w^w(\mu)}{p^w(x)} \right] \mathcal{N}(x; \mu, \Sigma) d\mu$$

$$= -\Sigma^{-1} \int \mu \left[ \frac{w^{gt}(\mu)}{p^{gt}(x)} \cdot \left( 1 + \sum_{k=1}^K B_s^k(\mu, x) \right) - \frac{w^{gt}(x)(\mu)}{p^{gt}(x)} \cdot \left( 1 + \sum_{k=1}^K B_w^k(\mu, x) \right) \right] \mathcal{N}(x; \mu, \Sigma) d\mu$$

$$= -\frac{\Sigma^{-1}}{p^{gt}(x)} \int \mu w^{gt}(\mu) \left( \sum_{k=1}^K \left[ B_s^k(\mu, x) - B_w^k(\mu, x) \right] \right) \mathcal{N}(x; \mu, \Sigma) d\mu$$

$$= \frac{\Sigma^{-1}}{p^{gt}(x)} \int \mu w^{gt}(\mu) \left( \sum_{k=1}^K \left[ \frac{B_w^k(\mu, x)}{B_s^k(\mu, x)} - 1 \right] B_s^k(\mu, x) \right) \mathcal{N}(x; \mu, \Sigma) d\mu$$

$$(21)$$

*Therefore, according to Equation (20) and Equation (21), we want $\Delta_1(x) \approx \Delta_2(x)$, i.e., $\|\Delta_1 - \Delta_2\|$ should be as small as possible. And the condition $\Delta_1(x) \approx \Delta_2(x)$ requires:*

$$\frac{B_w^k(\mu, x)}{B_s^k(\mu, x)} - 1 \approx 1. \tag{22}$$

*For any $\epsilon > 0$, under condition*

$$\left| \frac{B_w^k(\mu, x)}{B_s^k(\mu, x)} - 2 \right| \leq \epsilon \quad \text{for all } k = 1, \ldots, K, \mu \in \mathbb{R}^d, x \in \mathcal{X}, \tag{23}$$

we can compute the difference $\|\Delta_1(x) - \Delta_2(x)\|$ as

$$\|\Delta_1(x) - \Delta_2(x)\| \leq \frac{\|\Sigma^{-1}\|}{p^{gt}(x)} \left\| \int \mu w^{gt}(\mu) \left( \sum_{k=1}^K \left[ \frac{B_w^k(\mu, x)}{B_s^k(\mu, x)} - 2 \right] B_s^k(\mu, x) \right) \mathcal{N}(x; \mu, \Sigma) d\mu \right\|$$

$$\leq \frac{\|\Sigma^{-1}\|}{p^{gt}(x)} \int \|\mu\| w^{gt}(\mu) \left| \sum_{k=1}^K \left[ \frac{B_w^k(\mu, x)}{B_s^k(\mu, x)} - 2 \right] B_s^k(\mu, x) \right| \mathcal{N}(x; \mu, \Sigma) d\mu$$

$$\leq \epsilon \cdot \frac{\|\Sigma^{-1}\|}{p^{gt}(x)} \int \|\mu\| w^{gt}(\mu) \left( \sum_{k=1}^K |B_s^k(\mu, x)| \right) \mathcal{N}(x; \mu, \Sigma) d\mu. \tag{24}$$

Substituting yields

$$\|\Delta_1(x) - \Delta_2(x)\| \leq \epsilon \cdot \frac{\|\Sigma^{-1}\|}{p^{gt}(x)} \int \|\mu\| w^{gt}(\mu) \left( \sum_{k=1}^K |B_s^k(\mu, x)| \right) \mathcal{N}(x; \mu, \Sigma) d\mu. \tag{25}$$

Let:

$$C(x) = \frac{\int \|\mu\| w^{gt}(\mu) \left( \sum_{k=1}^K |B_s^k(\mu, x)| \right) \mathcal{N}(x; \mu, \Sigma) d\mu}{\left\| \int \mu w^{gt}(\mu) \left( \sum_{k=1}^K B_s^k(\mu, x) \right) \mathcal{N}(x; \mu, \Sigma) d\mu \right\|}. \tag{26}$$

Then:

$$\|\Delta_1(x) - \Delta_2(x)\| \leq \epsilon \cdot \frac{\|\Sigma^{-1}\|}{p^{gt}(x)} \int \|\mu\| w^{gt}(\mu) \left( \sum_{k=1}^K |B_s^k(\mu, x)| \right) \mathcal{N}(x; \mu, \Sigma) d\mu$$

$$= \epsilon \cdot \frac{\|\Sigma^{-1}\|}{p^{gt}(x)} \cdot C(x) \cdot \left\| \int \mu w^{gt}(\mu) \left( \sum_{k=1}^K B_s^k(\mu, x) \right) \mathcal{N}(x; \mu, \Sigma) d\mu \right\|$$

$$\leq \epsilon \cdot C(x) \cdot \|\Delta_2(x)\|. \tag{27}$$

The constant $C(x)$ serves as a **critical efficiency ratio** that quantifies the relationship between the cumulative magnitude of model deviations and their net directional effect on the score function. Mathematically, it represents the worst-case amplification factor between the absolute sum of strong model biases $|B_s^k(\mu, x)|$ across all Gaussian components and the resultant vector norm of their integrated effect. A value of $C(x) \approx 1$ indicates coherent directional biases that efficiently translate into score modifications, whereas $C(x) \gg 1$ signifies significant cancellation effects where substantial individual deviations yield minimal net impact on $\Delta_2(x)$. This ratio fundamentally determines the tightness of the error bound and characterizes the spatial effectiveness of the weak-to-strong generalization across the data manifold.

### A.3 GLOBAL ERROR REDUCTION VIA FISHER DIVERGENCE

Under the conditions of Theorems 1 and 2, in this section, we demonstrate the theoretical advantage of W2SD over the standalone Strong Model.

While deriving a tight global error bound for the entire sampling trajectory of generative diffusion models remains a challenging open problem in the field, we can rigorously demonstrate that W2SD achieves a lower local score estimation error compared to the strong model under the conditions derived in Theorem 2. Since the global sampling error is the accumulation of local discretization errors, reducing the local error directly contributes to a lower total error. We formalize this claim in Theorem 3, with the accompanying proof provided in Proof 3.

**Theorem 3 (Global Error Reduction via Fisher Divergence)** *Let $p^{\mathrm{gt}}(x)$ denote the ground truth data distribution. Let $\mathcal{J}(p^{\mathrm{gt}} \| p^\theta) = \frac{1}{2}\mathbb{E}_{x \sim p^{\mathrm{gt}}}[\|\nabla_x \log p^{\mathrm{gt}}(x) - \nabla_x \log p^\theta(x)\|^2]$ be the Fisher Divergence measuring the total score matching error.*

*Consider the refined score estimate of W2SD defined as $s_{\mathrm{w2sd}}(x) = \nabla_x \log p^{\mathrm{s}}(x) + \Delta_1(x)$. Under the approximation condition established in Theorem 2, where the point-wise estimation error is bounded by a factor $\epsilon \in [0, 1)$ such that $\|\Delta_1(x) - \Delta_2(x)\| \leq \epsilon \|\Delta_2(x)\|$, the total error of W2SD is strictly bounded by the total error of the strong model:*

$$\mathcal{J}(p^{\mathrm{gt}} \| p^{\mathrm{w2sd}}) \leq \epsilon^2 \mathcal{J}(p^{\mathrm{gt}} \| p^{\mathrm{s}}) < \mathcal{J}(p^{\mathrm{gt}} \| p^{\mathrm{s}}). \tag{28}$$

*This implies that W2SD theoretically achieves a globally superior generation quality compared to the standalone strong model.*

**Proof 3** *The Fisher Divergence for the strong model $\mathcal{M}^{\mathrm{s}}$ is given by the expectation of the squared norm of the strong-to-ideal gap $\Delta_2(x)$:*

$$\mathcal{J}(p^{\mathrm{gt}} \| p^{\mathrm{s}}) = \frac{1}{2} \int p^{\mathrm{gt}}(x) \|\nabla_x \log p^{\mathrm{gt}}(x) - \nabla_x \log p^{\mathrm{s}}(x)\|^2 dx = \frac{1}{2} \int p^{\mathrm{gt}}(x) \|\Delta_2(x)\|^2 dx. \tag{29}$$

*For the W2SD framework, the refined score is $s_{\mathrm{w2sd}}(x) = \nabla_x \log p^{\mathrm{s}}(x) + \Delta_1(x)$. The corresponding Fisher Divergence is:*

$$\begin{aligned}
\mathcal{J}(p^{\mathrm{gt}} \| p^{\mathrm{w2sd}}) &= \frac{1}{2} \int p^{\mathrm{gt}}(x) \|\nabla_x \log p^{\mathrm{gt}}(x) - s_{\mathrm{w2sd}}(x)\|^2 dx \\
&= \frac{1}{2} \int p^{\mathrm{gt}}(x) \|\nabla_x \log p^{\mathrm{gt}}(x) - (\nabla_x \log p^{\mathrm{s}}(x) + \Delta_1(x))\|^2 dx \\
&= \frac{1}{2} \int p^{\mathrm{gt}}(x) \|(\nabla_x \log p^{\mathrm{gt}}(x) - \nabla_x \log p^{\mathrm{s}}(x)) - \Delta_1(x)\|^2 dx \\
&= \frac{1}{2} \int p^{\mathrm{gt}}(x) \|\Delta_2(x) - \Delta_1(x)\|^2 dx.
\end{aligned} \tag{30}$$

*Based on the premise derived from Theorem 2 (relative bias consistency), we have the point-wise bound $\|\Delta_2(x) - \Delta_1(x)\| \leq \epsilon\|\Delta_2(x)\|$. Substituting this inequality into the integral:*

$$
\begin{aligned}
\mathcal{J}(p^{\text{gt}}\|p^{\text{w2sd}}) &\leq \frac{1}{2}\int p^{\text{gt}}(x)\,(\epsilon\|\Delta_2(x)\|)^2\,dx \\
&= \epsilon^2 \cdot \left(\frac{1}{2}\int p^{\text{gt}}(x)\|\Delta_2(x)\|^2 dx\right) \\
&= \epsilon^2 \mathcal{J}(p^{\text{gt}}\|p^{\text{s}}).
\end{aligned}
\tag{31}
$$

*Since $0 \leq \epsilon < 1$ holds for well-aligned weak-to-strong pairs (as discussed in Theorem 2), it follows that $\mathcal{J}(p^{\text{gt}}\|p^{\text{w2sd}}) < \mathcal{J}(p^{\text{gt}}\|p^{\text{s}})$.*

## B  RELATED WORK

In this section, we review existing works relevant to W2SD.

**Weak-to-Strong Mechanism**   The concept of improving weak models into strong models originates from the AdaBoost (Høgsgaard et al., 2023), which constructs a more accurate classifier by aggregating multiple weak classifiers. Building upon this, recent theoretical advances (Green Larsen & Ritzert, 2022; Høgsgaard et al., 2023) introduced a provably optimal weak-to-strong learner, establishing a robust theoretical foundation for this weak-to strong paradigm. In the field of LLMs, several studies (Chen et al.; Burns et al., 2023) have utilized weak models as supervisory signals to facilitate the alignment of LLMs. This paradigm of weak-to-strong generation during training has similarly been investigated in the context of diffusion model training (Chen et al., 2025). The research by (Sugiyama et al., 2013) recommended directly estimating the density gap between weak and strong models instead of separate estimations. And Auto-guidance (Karras et al., 2024) examines the mechanism of CFG and achieves improved results through interpolation-based perturbation. In contrast, W2SD utilizes the reflection by leveraging the gap between denoising and inversion to steer generation towards user-defined directions. And we note that W2SD generalizes the concept of ideal model, with experiments across human preference, personalization, fidelity and so on, confirming its wide applicability.

**Diffusion Inference Enhancement**   The study of inference scaling laws in diffusion models has recently become a prominent focus within the research community (Ma et al., 2025; Ye et al.; Liu et al., 2024a; Li et al., 2026; Shao et al., 2024b; Xie et al.). This line of work can be traced back to Re-Sampling (Lugmayr et al., 2022), which iteratively refines latent variables by injecting random Gaussian noise, effectively reverting the noise level to a previous scale. This iterative paradigm has been utilized in subsequent works, including universal conditional control (Bansal et al., 2023), video generation (Wu et al., 2023a), and protein design (Jumper et al., 2021), to enhance inference performance. However, it has primarily been treated as a heuristic trick, with its underlying mechanisms remaining underexplored. Z-Sampling (Bai et al., 2024) extended this paradigm by replacing random noise injection with inversion operations and identified the guidance gap between denoising and inversion as a critical factor. This phenomenon has also been validated in subsequent studies (Zhou et al., 2024; Shao et al., 2024a; Ahn et al., 2024). In our work, we systematically unify these inference enhancement methods, demonstrating that their essence lies in approximating the strong-to-ideal gap via the weak-to-strong gap, and integrate them into a unified reflective framework, W2SD, through theoretical and empirical analysis.

**Reflection Diffusion Model**   Large language models (LLMs) and multimodal large language models (MLLMs) exhibit the emergent ability to simulate human cognitive processes through reflective reasoning (Cheng et al., 2025; Renze & Guven, 2024; Shinn et al., 2023; Shao et al., 2025b), building upon their foundational comprehension and generative competencies. Recent works (Guo et al., 2025; Bai et al., 2024; Lou & Ermon, 2023; Zhuo et al., 2025) have increasingly focused on integrating the concept of reflection into diffusion models. Z-Sampling (Bai et al., 2024) effectively injects semantic meaning from prompts into the generation process through reflective operations, thereby significantly enhancing prompt alignment. (Pan et al., 2025) effectively incorporates reflective reasoning into reinforcement learning, equipping diffusion models with advanced reasoning capacities. And (Guo et al., 2025) pioneered the introduction of reflection mechanisms in autoregressive image

generation for self-correcting unsatisfactory generated images. Particularly, the reflection mechanism has also been applied to image editing in works such as (Jiao et al., 2025) on UnifiedEdit. The "reflected diffusion model" proposed by (Lou & Ermon, 2023), reduces numerical errors via reflected stochastic differential equations, while W2SD achieves targeted distribution shifts through dual-model collaboration.

## C  EXPERIMENT SETTINGS

In this section, we introduce the details of hyperparameters, metrics and datasets used in the experiments.

### C.1  HYPERPARAMETERS

**Weight Gap**    For the W2SD based on weight gap, we consider full-parameter tuning, LoRA-based efficient tuning, and the MoE mechanism.

For the full-parameter tuning, we first set SD1.5 (Rombach et al., 2022) as the weak model and the fine-tuned DreamShaper v8 as the strong model, which demonstrates superior performance in terms of human preference and achieves high-quality generation results. Similarly, we also use SDXL (Podell et al.) as the weak model and Juggernaut-XL as the strong model to further evaluate W2SD's performance under weight gaps.

For the efficient tuning, we distinguish strong and weak models by adjusting the LoRA scale. First, we use the xlMoreArtFullV1 LoRA checkpoint to test W2SD's ability to enhance overall image quality. Additionally, to validate the ability of W2SD to improve personalization, we select a series of personalized LoRAs, as detailed in Appendix C.3. For the strong model, the LoRA scale is set to 0.8, while for the weak model, it is set to -1.5.

Following the default settings (Bai et al., 2024), we set the denoising steps $T = 50$, and the guidance scale to 5.5. Reflection operations steps $\lambda = T - 1$. Notably, to eliminate influence from guidance gaps, the guidance scale of $M^s$ and $M^w$ are both set to 1.0 during reflection, ensuring the guidance gap is zero.

For the MoE (Mixture of Experts) mechanism, we select DiT-MoE-S (Fei et al., 2024) as the strong model, which routes the top 2 optimal experts out of 8 during the inference process. The weak model, in contrast, is configured to route the 2 least optimal experts out of 8. Following the default settings, the denoising steps $T = 50$, and the guidance scale is set to 1.5.

**Condition Gap**    In the W2SD research based on condition gaps, we focus on analyzing the gaps caused by two mechanisms: guidance scale and prompt semantics.

By adjusting the guidance scale to distinguish the strong and weak model, we adapt the same settings as Z-Sampling (Bai et al., 2024): the guidance scale of $\mathcal{M}^s$ was set to 5.5, and the guidance scale of $\mathcal{M}^w$ is set to 0. The diffusion model used is SDXL.

For semantic gaps, we set $\mathcal{M}^w$ to use the GenEval prompt, which is short (4-5 words), ambiguous, and coarse, often resulting in uncontrolled outputs. In contrast, $\mathcal{M}^s$ uses refined prompts enhanced by QWen-Plus (Yang et al., 2024), providing greater detail and semantic richness. During the reflection process, the guidance scale was set to 1.0 to eliminate the influence of guidance gaps on the results.

Similar to the weight gap setup, we set $T = 50$, $\lambda = T - 1$, and the denoising guidance scale to 5.5.

**Sampling Pipeline Gap**    We demonstrate that W2SD can also generalize to capability gaps across different diffusion pipelines. Specifically, we select ControlNet (Zhang et al., 2023) as $\mathcal{M}^s$, with the control scale set to the default value of 1.0. The standard sampling pipeline (Song et al., 2020) is chosen as the weak model. Consistent with the weight gap setup, we configure $T = 50$, $\lambda = T - 1$, and guidance scale of 5.5. During reflection operation, the guidance scale for both $\mathcal{M}^s$ and $\mathcal{M}^w$ are set to 0.5 to eliminate the influence of guidance gaps.

## C.2 METRICS

**AES.**  AES (Schuhmann et al., 2022) is an evaluation metric that assesses the visual quality of generated images by analyzing key aesthetic attributes such as contrast, composition, color, and detail, thereby measuring their alignment with human aesthetic standards.

**PickScore.**  PickScore (Kirstain et al., 2023) is a CLIP-based metric model trained on a comprehensive open dataset containing text-to-image prompts and corresponding real user preferences for generated images, specifically designed for predicting human aesthetic preferences.

**HPS v2.**  Building upon the Human Preference Dataset v2 (HPD v2), a comprehensive collection of 798,090 human preference judgments across 433,760 image pairs, (Wu et al., 2023b) developed HPS v2 (Human Preference Score v2) through CLIP fine-tuning, establishing a more accurate predictive model for human preferences in generated images.

**MPS.**  MPS (Zhang et al., 2024) is a metric for text-to-image model evaluation, trained on the MHP dataset containing 918,315 human preference annotations across 607,541 images. This novel metric demonstrates superior performance by effectively capturing human judgments across four critical dimensions: aesthetic quality, semantic alignment, detail fidelity, and overall assessment.

## C.3 DATASETS

**Pick-a-Pic.**  The Pick-a-Pic dataset (Kirstain et al., 2023), collected through user interactions with a dedicated web application for text-to-image generation, systematically records each comparison with a prompt, two generated images, and a preference label (indicating either a preferred image or a tie when no significant preference exists). Following Z-Sampling (Bai et al., 2024), we utilize the initial 100 prompts as a representative test set, which provides adequate coverage to assess model performance.

**Drawbench.**  DrawBench (Saharia et al., 2022) is a comprehensive evaluation benchmark for text-to-image models, featuring approximately 200 text prompts across 11 distinct categories that assess critical capabilities including color rendering, object counting, and text generation.

**GenEval.**  Geneval (Ghosh et al., 2024) is an object-focused framework that evaluates image composition through object co-occurrence, position, count, and color. Using 553 prompts, it achieves 83% agreement with human judgments on image correctness.

**VBench**  VBench (Huang et al., 2024) is a comprehensive benchmark for video generation models, featuring a hierarchical evaluation framework across multiple quality dimensions. It supports both automatic and human assessment, with VBench++ extending to text-to-video and image-to-video tasks while incorporating trustworthiness evaluation.

**Peronalization Dataset**  To evaluate the performance gains of W2SD in personalized generation, we selected 20 LoRA checkpoints from the Civitai platform, covering a diverse range of categories, including persons (e.g. Anne Hathaway, Scarlett Johansson), animals (e.g., Scottish Fold cat, prehistoric dinosaur), styles (e.g., Disney style, parchment style), anime characters (e.g. Sun Wukong, Bulma) and objects (e.g. cars).

# D  SUPPLEMENTARY EXPERIMENTAL RESULTS

In this section, we present more quantitative and qualitative results of W2SD.

## D.1 QUANTITATIVE RESULTS

**Results of W2SD in other benchmarks**  To further validate the effectiveness of W2SD, we also conducted experiments on Drawbench (Saharia et al., 2022). For clarity, we have systematically organized and summarized these experiments in Table 1. In Drawbench, we report results based

Table 8: Quantitative results of W2SD based on a full parameter fine-tuning strategy. Our framework generates results better aligned with human preferences. Datasets: Drawbench.

| Method | HPS v2 ↑ | AES ↑ | PickScore ↑ | MPS ↑ |
|---|---|---|---|---|
| SD1.5 | 25.3601 | 5.2023 | 21.0519 | - |
| DreamShaper | 28.7845 | 5.7047 | 21.8522 | 47.8813 |
| **W2SD** | **28.7901** | **5.7847** | **21.9057** | **52.1192** |
| SDXL | 28.5536 | **5.4946** | 22.2464 | - |
| Juggernaut-XL | 28.9085 | 5.3455 | 22.4906 | 47.5648 |
| **W2SD** | **29.3246** | 5.4261 | **22.5803** | **52.4358** |

Table 9: Quantitative results of W2SD based on guidance gap. Model: SDXL. Datasets: DrawBench.

| Method | HPS v2 ↑ | AES ↑ | PickScore ↑ | MPS ↑ |
|---|---|---|---|---|
| SD1.5 | 25.3601 | 5.2023 | 21.0519 | 47.7075 |
| **W2SD** | **25.8234** | **5.2157** | **21.2079** | **52.2934** |
| SDXL | 28.5536 | **5.4946** | 22.2464 | 40.9590 |
| **W2SD** | **30.1426** | **5.6600** | **22.4434** | **59.0415** |

Table 10: Quantitative results of W2SD based on human preference LoRA model. Our framework generates results better aligned with human preferences. Datasets: Pick-a-Pic.

| Method | HPS v2 ↑ | AES ↑ | PickScore ↑ | MPS ↑ |
|---|---|---|---|---|
| SD1.5 | 24.9558 | 5.5003 | 20.1368 | - |
| Dpo-Lora | 25.5678 | 5.5804 | 20.3514 | 44.2889 |
| **W2SD** | **26.0825** | **5.6567** | **20.5096** | **55.7106** |
| SDXL | 29.8701 | 6.0939 | 21.6487 | - |
| xlMoreArtFullV1 | 32.8040 | 6.1176 | 22.3259 | 48.2224 |
| **W2SD** | **33.5959** | **6.2252** | **22.3644** | **51.7770** |

Table 11: Quantitative results of W2SD based on human preference LoRA model. Our framework generates results better aligned with human preferences. Datasets: DrawBench.

| Method | HPS v2 ↑ | AES ↑ | PickScore ↑ | MPS ↑ |
|---|---|---|---|---|
| SD1.5 | 25.3601 | 5.2023 | 21.0519 | - |
| Dpo-Lora | 25.8896 | 5.2895 | 21.2308 | 49.3617 |
| **W2SD** | **25.9431** | **5.3553** | **21.2589** | **50.6399** |
| SDXL | 28.5536 | 5.4946 | 22.2464 | - |
| xlMoreArtFullV1 | 31.2727 | 5.5487 | 22.7721 | 47.0396 |
| **W2SD** | **32.34857** | **5.7595** | **22.8301** | **52.9588** |

on weight gap (see Tables 8 and 11) and guidance gap (see Table 9). Additionally, in Pick-a-Pic, we report results based on weight gap (see Table 10). Notably, W2SD demonstrates consistent improvements across all evaluated metrics.

**Results of W2SD in Video Generation Task**   We also validate the performance of W2SD on video generation task to demonstrate its broad applicability. We randomly select 200 prompts from VBench (Huang et al., 2024) as test prompt cases and focus on analyzing the AnimateDiff (Guo et al.) as video generation model.

Table 12: Quantitative results of W2SD on the video generation task. Model: AnimateDiff. Datasets: VBench.

| Method | Subject Consistency ↑ | Background Consistency ↑ | Motion Smoothness ↑ | Dynamic Degree ↑ | Aesthetic Quality ↑ | Image Quality ↑ |
|---|---|---|---|---|---|---|
| AnimateDiff (SDXL) | 91.4152% | 95.8491% | 94.1647% | **45.0000%** | 55.3108% | 58.4293% |
| AnimateDiff (Juggernaut-XL) | 96.0820% | 97.5463% | **96.8834%** | 13.0653% | 57.7097% | 64.1674% |
| **W2SD** | **97.1398%** | **97.9386%** | 96.8706% | 8.0000% | **59.3736%** | **64.6987%** |

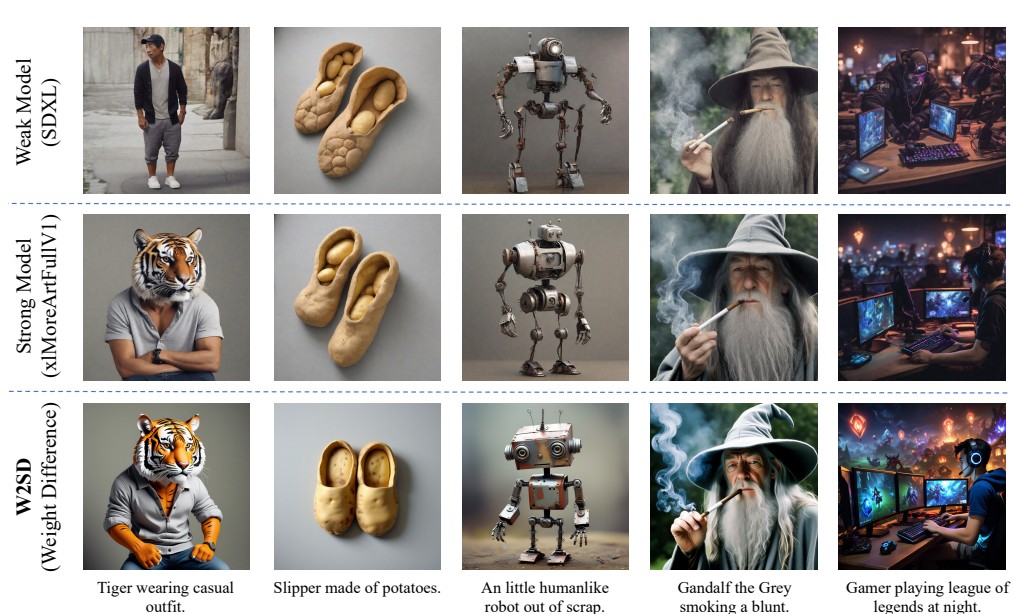

Figure 15: Qualitative results of W2SD based on weight gaps (human preference). Here we select xlMoreArtFullV1 as the strong model and SDXL as the weak model. W2SD can effectively enhance the performance of human preference.

For the strong model $\mathcal{M}^{\mathrm{s}}$, we employ AnimateDiff with Juggernaut-XL using guidance scale of 3.0, while the weak model $\mathcal{M}^{\mathrm{w}}$ utilizes AnimateDiff with SDXL at guidance scale of 1.0. This setup introduces both weight and guidance gaps, meeting the conditions required by W2SD.

In Table 12, W2SD achieves significant improvements across different dimensions such as Subject Consistency, Background Consistency, Aesthetic Quality and mage Quality, confirming its effectiveness in video generation task. Notably, in the Motion Smoothness dimension, W2SD exhibits a performance degradation due to Juggernaut-XL's inherent limitations compared to SDXL. This observation further validates our theory in Section 3.1.

## D.2 QUALITATIVE RESULTS

**Weight Gap**    In Figure 15, we present visualization results of W2SD based on weight gap. Specifically, to enhance human preference for generated images, we select xlMoreArtFullV1 as the strong model and SDXL as the weak model.

Additionally, by setting the LoRA-based personalized model as strong model and the un-finetuned base model as weak model, Figure 17 showcases the improvement of W2SD in personalized generation effectiveness.

**Condition Gap**    In Figure 16, we present the visualization results of W2SD based on condition gaps. When the strong model utilizes detailed and semantically rich prompts, while the weak model relies on simple prompts (containing only 4–5 words), W2SD effectively captures fine-grained conditional features. Notably, Z-Sampling (Bai et al., 2024) is a special case of W2SD based on guidance gaps, with extensive visual evidence already provided; thus, we omit additional visualizations here.

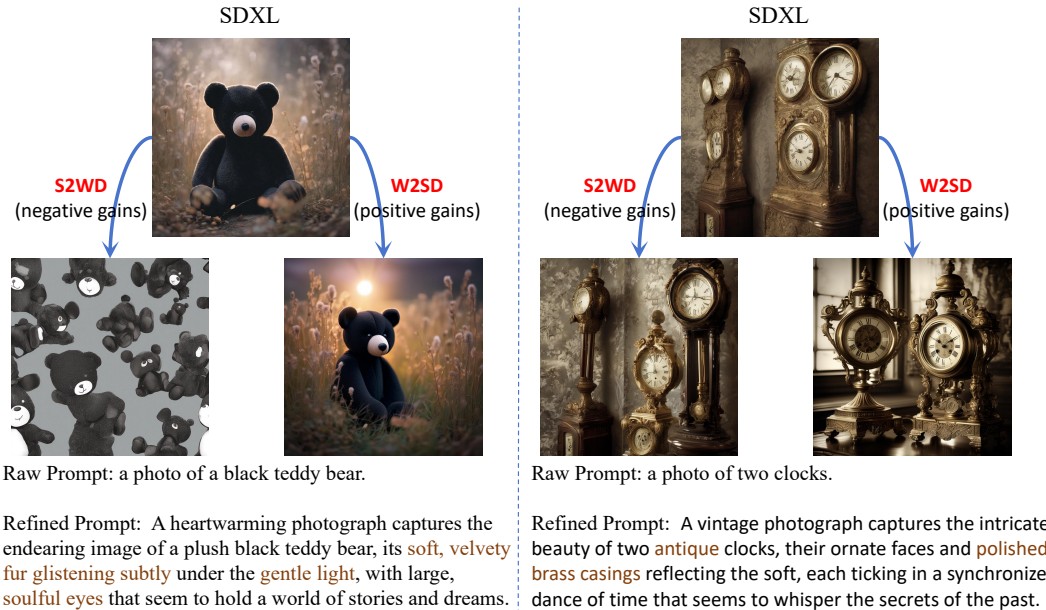

Figure 16: Qualitative results of W2SD based on semantic gaps between prompts, which refines the generation process by placing greater emphasis on the fine-grained details.

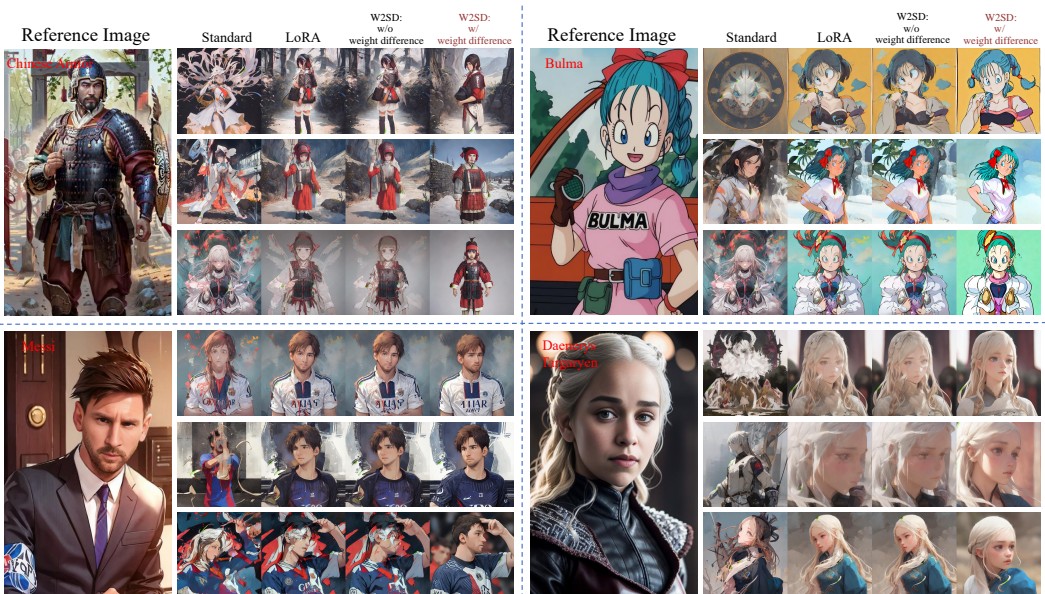

Figure 17: Qualitative results of W2SD based on weight gaps (personalization). Here we set LoRA-based personalized model as strong model and the standard model as weak model

**Sampling Pipeline Gap** In Figure 10, we present the visualization results of W2SD based on the gaps in the sampling pipeline. By incorporating reference image information through Ip-adapter during the denoising process and employing standard DDIM for inversion, W2SD ensures that the generated image adheres more closely to the given stylistic conditions, resulting in higher-quality results.

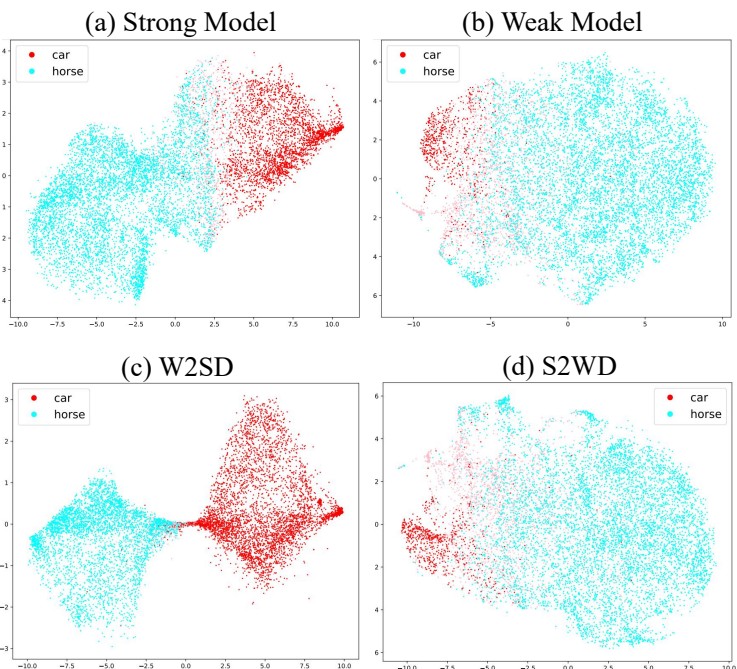

Figure 18: The CLIP feature corresponding to the generated image ($32 \times 32 \times 3$) is projected into a 2D space. W2SD effectively disentangles the representations of "car" and "horse". (a) $\mathcal{M}^s$ demonstrates the ability to generate cars; (b) $\mathcal{M}^w$ can hardly generate cars; (c) W2SD balances the generation distribution, increasing the likelihood of generating cars; (d) S2WD (i.e., $\mathcal{M}^s_{inv}(\mathcal{M}^w(\cdot))$) exacerbates the imbalance in data generation.

**W2SD Performance with Few Steps**   Here we also validate the performance of our method under different $\lambda$ and $T$ generation settings. In Figure 19, we choose **dreamshaper-xl-turbo** as the base model, using DPM Solver++ as the scheduler, and evaluate W2SD based on LoRA gap under few steps (T=8). In Figure 20, we utilize **SDXL** as the base model, using DDIM as the scheduler, and evaluate W2SD based on LoRA gap under multi steps (T=20).

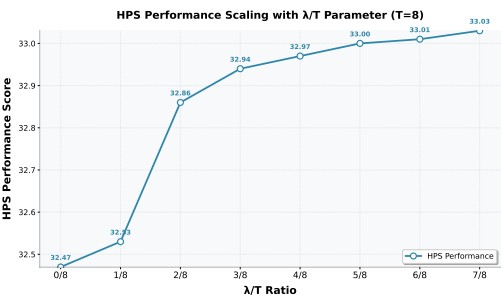
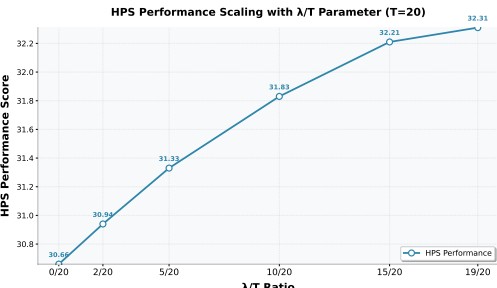

Figure 19: HPS Performance Scaling with $\lambda/T$ Parameter ($T = 8$. BaseModel:**dreamshaper-xl-turbo**. Scheduler: DPM Solver++)

Figure 20: HPS Performance Scaling with $\lambda/T$ Parameter ($T = 20$). BaseModel:**SDXL**. Scheduler: DDIM)

For further analysis, we visualize the denoising trajectories under both **few-step (5-step)** and **multi-step (20-step)** settings in a one-dimensional case, as shown in Figure 21. It can be observed that the iterative method alternating between weak and strong models achieves effective results in both few-step and multi-step scenarios.

One notable observation is that in the 5-step case (top-left subfigure in Figure 21), the trajectory of W2S appears less smooth. Nevertheless, due to the macroscopic consistency in the directional discrepancy between the weak and strong models, the optimized trajectory consistently aligns toward a broadly shared optimization objective.

This finding further reveals that even under few-step conditions, the macro-level consistent guidance signal provided by W2SD can effectively steer the sampling trajectory toward more desirable regions in the image distribution.

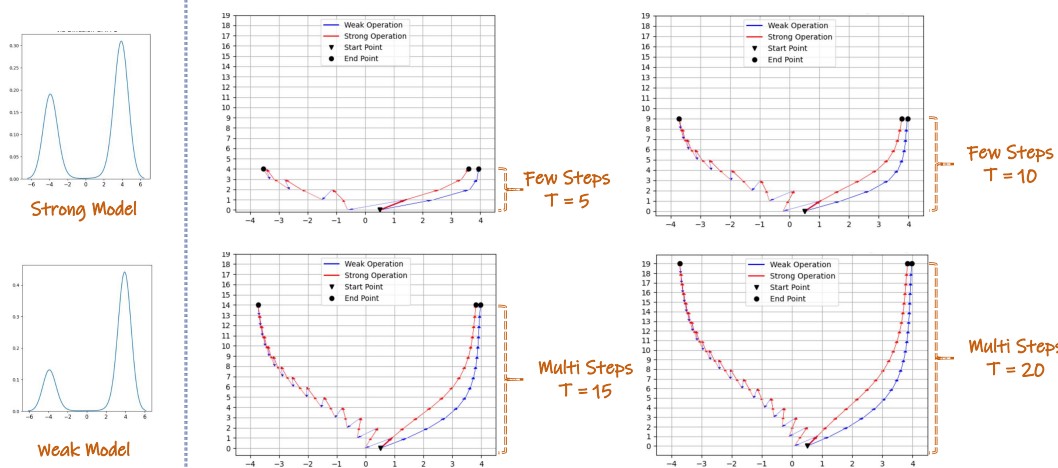

Figure 21: W2SD Ensures Effective Guidance Across Sampling Budgets: While 5-step trajectories (top-left) appear less smooth, the macroscopic consistency of guidance steers the sampling process effectively.

**Analysis of the Limitations and Failure Scenarios in W2SD** In Theorem 2, we demonstrate that the weak-to-strong gap cannot consistently serve as a reliable proxy for the strong-to-ideal gap. This limitation manifests in two typical scenarios: model conflict and model over-similarity. Representative examples of these failure modes are illustrated in Figure 22 and Figure 23, respectively.

Taking Figure 22 as an example, here $M_1$ represents a model associated with the "Disney Style", while $M_2$ corresponds to the "Ink Style". Our goal was to generate a Disney-style dog in ink wash painting; however, this objective could not be achieved regardless of whether $M_1$ or $M_2$ was used as the strong model. This failure stems from the fact that such a model pair is not well-defined for the task of generating "**Disney-style ink paintings**."

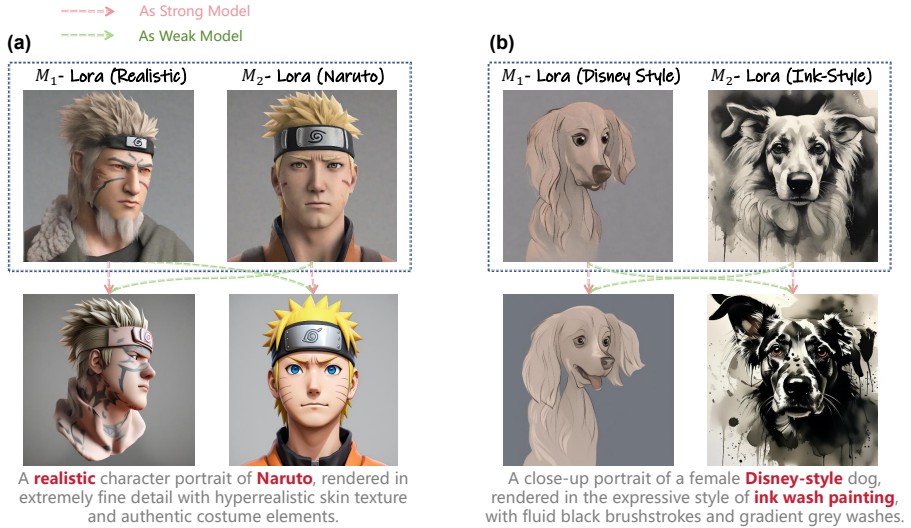

Figure 22: Failure Scenario 1: Model Conflict. W2SD fails when weak and strong models have mutually exclusive features.

Moreover, when the capabilities of the two models in the model pair are too similar, the resulting model gap becomes too weak and lacks a clear directional signal. In such cases, W2S fails to provide meaningful performance gains. In Figure 23, we selected two similar preference LoRAs (specifically xlMoreArtFullV1 and styleLoraRealis). Due to the marginal capability gap between them, the optimization effect remains notably weak and highly inconsistent—while some cases (e.g., bottom-right in Figure 23) show slight improvement, others (e.g., top-right) exhibit clear performance degradation.

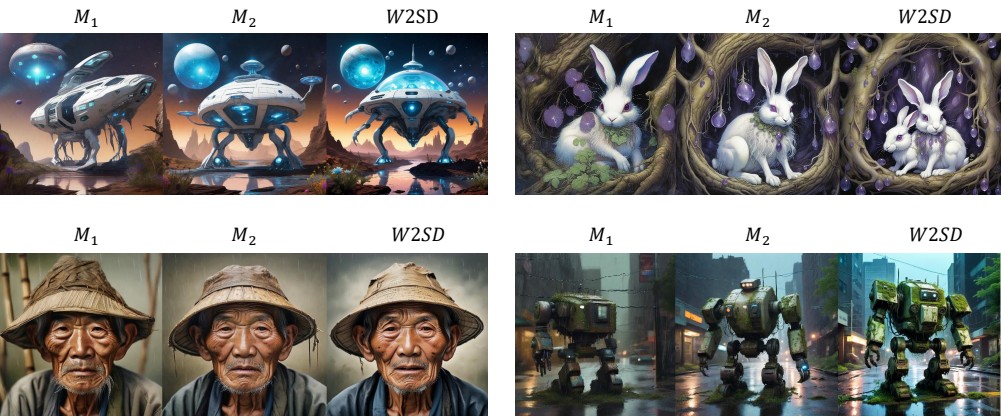

Figure 23: Failure Scenario 2: Model Over-Similarity. W2SD yields limited gains when weak and strong models possess highly similar features, resulting in a weak guidance signal. Here, $M_1$ (strong model) corresponds to xlMoreArtFullV1, while $M_2$ (weak model) corresponds to styleLoraRealis.

**Analysis of the Reward Model-Based W2S Variant**  We now present a variant of the W2S approach (Please see Algorithm 2) which leverages a reward model (e.g., HPS or PickScore) to automatically designate the strong and weak models at each step. This variant addresses the key limitation of fixed model pairing by introducing a dynamic selection mechanism. At each denoising step, the reward model performs real-time assessment of the intermediate outputs from both models. The model that generates the higher-scoring output is automatically designated as the 'strong" model for that particular step, while the other serves as the "weak" model.

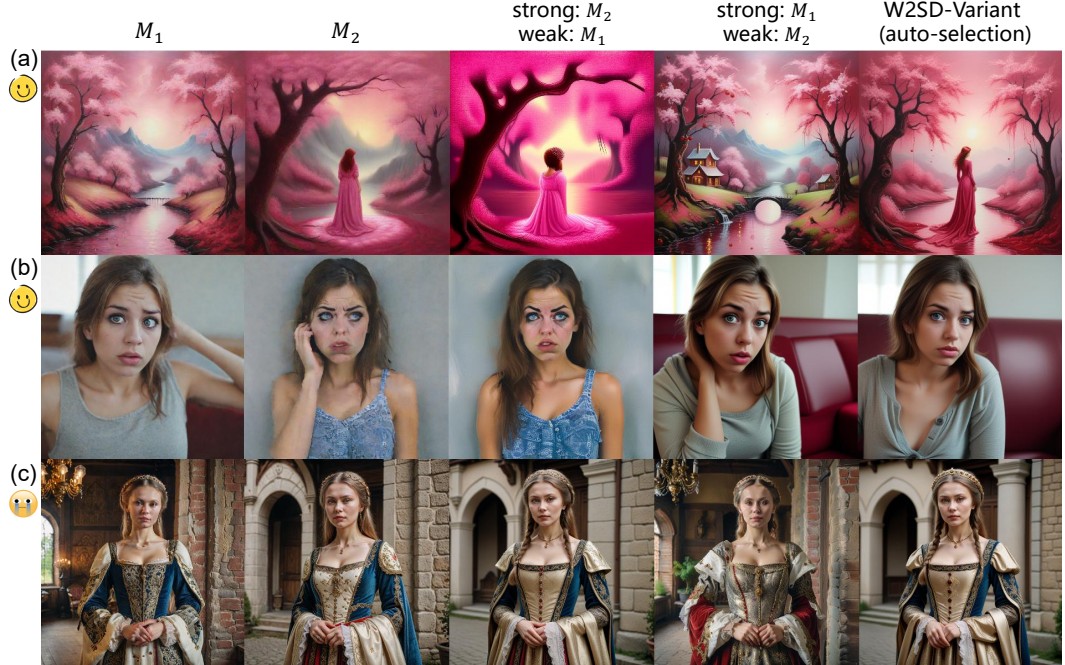

Figure 24: W2SD auto-selection provides flexibility by eliminating the need for pre-defined models (a, b), but fails when models are too similar to offer meaningful guidance (c).

To validate this perspective, we experiment with two related, preference-aligned LoRA weights[2] ($M^1$, $M^2$). We compare the standard W2SD (Algorithm 1) against the dynamic variant (Algorithm 2) on the same model set under identical conditions, with results shown in Table 13. We present the visualization results in Figure 24.

---

**Algorithm 2** W2S with Dynamic Model Selection

---

1: **Input:** Model $\mathcal{M}_1$, Model $\mathcal{M}_2$, Total Steps: $T$, Evaluation Metric: $\mathcal{E}$ (e.g., HPS, PickScore), Threshold $\epsilon_r$
2: **Output:** Refined Data $\mathbf{x}_0$
3: Initialize $\mathbf{x}_T$ {Sample from Gaussian noise}
4: **for** $t = T$ **to** 1 **do**
5:     # Dynamic Model Selection
6:     $\mathbf{c}_1 \leftarrow \mathcal{M}^1(\mathbf{x}_t, t), \mathbf{c}^2 \leftarrow \mathcal{M}^2(\mathbf{x}_t, t)$
7:     $r_1 \leftarrow \mathcal{E}(\mathbf{c}_1), r_2 \leftarrow \mathcal{E}(\mathbf{c}_2)$
8:     **if** $r_1 > r_2$ **and** $|r_1 - r_2| > \epsilon_r$ **then**
9:         $\mathcal{M}^s \leftarrow \mathcal{M}^1, \mathcal{M}^w \leftarrow \mathcal{M}^2$
10:     **else**
11:         $\mathcal{M}^s \leftarrow \mathcal{M}^2, \mathcal{M}^w \leftarrow \mathcal{M}^1$
12:     **end if**
13:     # W2SD with Reflection
14:     $\tilde{\mathbf{x}}_t \leftarrow \mathcal{M}^w_{\text{inv}}(\mathcal{M}^s(\mathbf{x}_t, t), t)$
15:     $\mathbf{x}_{t-1} \leftarrow \mathcal{M}^s(\tilde{\mathbf{x}}_t, t)$
16: **end for**
17: **return** $\mathbf{x}_0$ {Final refined data}

---

[2]Specifically, **xlMoreArtFullV1** and **xlMoreArtFullBeta2** from Civitai.

Table 13: HPS v2 evaluation of individual models and W2SD variants. Experiments conducted on a curated set of 20 prompts from the Pick-a-Pic dataset.

| | $\mathcal{M}^1$ (Single) | $\mathcal{M}^2$ (Single) | $\mathcal{M}^1(\mathcal{M}^2_{inv})$ (W2SD) | $\mathcal{M}^2(\mathcal{M}^1_{inv})$ (W2SD) | **Auto-Selection** (W2SD) |
|---|---|---|---|---|---|
| HPS v2 ↑ | 30.16 | 26.66 | 27.16 | 30.29 | **30.89** |

## E MECHANISM ANALYSIS AND UNDERSTANDING

### E.1 W2SD IN CIFAR-10

In this subsection, we investigate the performance of W2SD on real image data. For ease of analysis, we select two classes from CIFAR-10 (Krizhevsky et al., 2009): **car** and **horse**. Specifically, we train the weak and strong diffusion models on distinct datasets. For $\mathcal{M}^s$, the dataset consists of 5,000 horse images and 2,500 car images. For $\mathcal{M}^w$, it comprises 5,000 horse images and 500 car images. In this scenario, due to the imbalance in training dataset, both $\mathcal{M}^w$ and $\mathcal{M}^s$ are more inclined to generate horses. Moreover, since $\mathcal{M}^w$ is trained on a limited number of car images, it rarely generates cars.

In Figure 7, we note that W2SD can help balance the image categories, enhancing inference process to increase the probability of generating cars. In Figure 18, we also perform a t-SNE dimensionality reduction (Van der Maaten & Hinton, 2008) on the CLIP features of the generated images. W2SD effectively disentangles the representations of "car" and "horse" in the 2D space. Notably, the ratio of "horse" to "car" under W2SD approaches 1:1. In contrast, applying the negative reflection operator $\mathcal{M}^s_{inv}(\mathcal{M}^w(\cdot))$ worsens the data imbalance, validating the effectiveness of our framework.

### E.2 VISUALIZATION ANALYSIS

To provide a more intuitive demonstration of the W2SD effect, we present more 1D visualization examples in Figure 25 to aid understanding. As can be seen, W2SD has a certain probability of "correcting" the denoising trajectory of the model, resulting in more balanced generation outcomes. The settings here are consistent with those in Figure 5.

We also empirically validate that the weak-to-strong gap $\Delta_2$ can effectively bridges the strong-to-ideal gap $\Delta_1$. We first compute **CosineSimilarity**$(\Delta_1(t), \Delta_2(t))$ at each timestep $t$. We train three models on CIFAR-10: the weak model (100 epochs), the strong model (300 epochs), and the ideal model (fully converged, 500 epochs). Here $\Delta_1(t) = \epsilon_{ideal}(t) - \epsilon_{strong}(t)$ and $\Delta_1(t) = \epsilon_{strong}(t) - \epsilon_{weak}(t)$. In the table below, the angle between $\Delta_1(t)$ and $\Delta_2(t)$ remains below $90^o$, confirming that weak-to-strong gap can reliably bridge strong-to-ideal gap.

To further validate this claim that $\Delta_2$ can bridge $\Delta_1$, we provide additional visualizations. As illustrated in Figure 27, the W2SD trajectory is visualized in 2D space, where its denoising direction (black) demonstrates closer alignment with the ideal direction (green). Moreover, the angles between the strong (red)-to-weak (blue) and ideal (green)-to-strong (red) vectors consistently remain within $90°$.

### E.3 THE IMPACT OF APPROXIMATION ERROR IN THE INVERSION PROCESS

In Section 2, we assume that the inversion and denoising process are reversible, meaning that for the same generative model (including the same guidance scale, etc.), $\mathcal{M}_{inv}(\mathcal{M}(x_t, t), t) = x_t$. However, in practice, the inversion process inevitably introduces errors. Specifically, for $x_t$ at time $t$, the inversion process actually used in the algorithm implementation is as

$$\tilde{x}_t = x_{t-\Delta t} - \sigma^{2t} s_\theta(x_{t-\Delta t}, t)\Delta t \tag{32}$$

$$= x_{t-\Delta t} - \sigma^{2t}(\underbrace{s_\theta(x_{t-\Delta t}, t) - s_\theta(x_t, t)}_{\text{Inversion Error } E_t} + s_\theta(x_t, t))\Delta t, \tag{33}$$

And the effect of W2S on the latent variable in Theorem 1 can also be expressed as

$$\tilde{x}_t = x_t + \sigma^{2t}\Delta t(\nabla_{x_t} \log p^s_t(x_t) - \nabla_{x_t} \log p^w_t(x_t) - E_t), \tag{34}$$

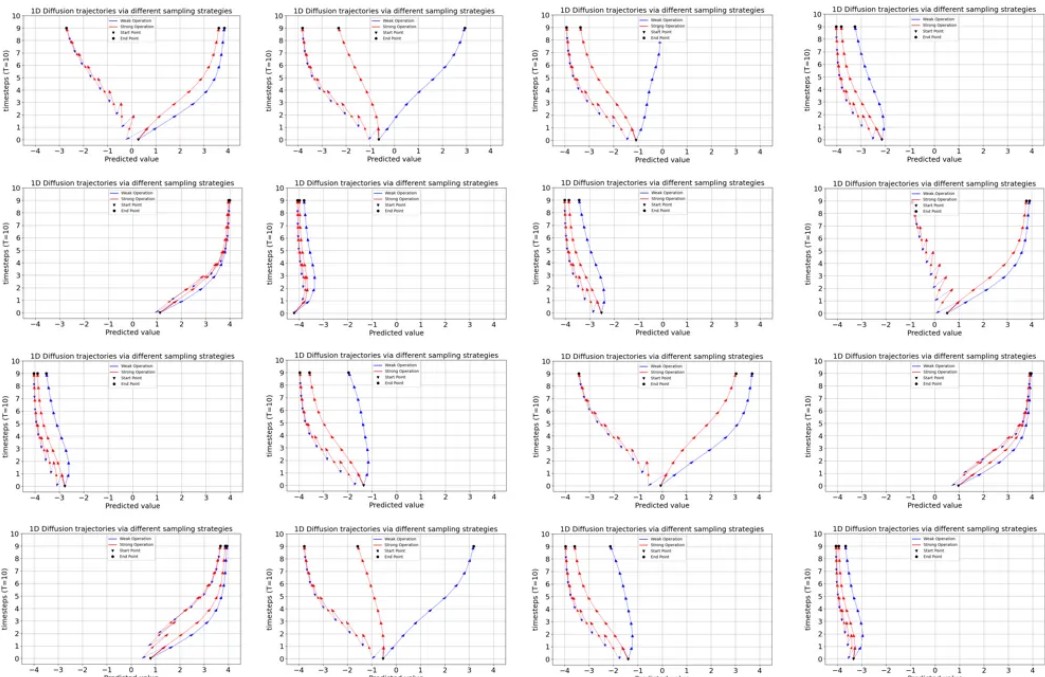

Figure 25: Denoising trajectories across different settings (1-D Gauss). The weak model (blue) generates only right-peak data due to missing left-peak training samples, while the strong model (red) produces data between both peaks. W2SD balances the distribution by leveraging the reflective operator $\mathcal{M}_{\text{inv}}^{\text{w}}(\mathcal{M}^{\text{s}}(\cdot))$.

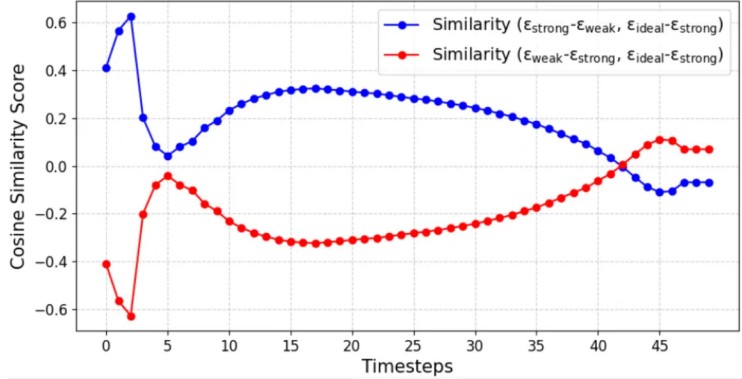

Figure 26: Positive cosine similarity between $\Delta_1$ and $\Delta_2$ ($\cos\Theta < 90°$).

where Inversion Error $E_t$ represents the approximation error in the inversion process at time $t$. Since $\Delta t$ is very small, $s_\theta(x_t, t)$ and $s_\theta(x_{t-\Delta t}, t)$ are assumed to be approximately equal by default, i.e., $E_t \approx 0$.

To further analyze the impact of $E_t$ on the W2SD, we simplify the analysis by replacing $E_t$ with Gaussian noise of controllable magnitude as

$$\tilde{x}_t = x_t + \sigma^{2t}\Delta t(\nabla_{x_t}\log p_t^{\text{s}}(x_t) - \nabla_{x_t}\log p_t^{\text{w}}(x_t) - k\epsilon). \tag{35}$$

By varying the value of $k$, we adjust the magnitude of the error $E_t$ to study its effect on the generated results. We analyze W2SD based on weight gaps, with the strong model using xlMoreArt-FullV1 and the weak model using SDXL.

In Table 14, as $k$ increases, indicating a larger approximation error $E_t$, the gains from W2SD diminish. This finding is consistent with Z-Sampling (Bai et al., 2024)'s results, demonstrating that

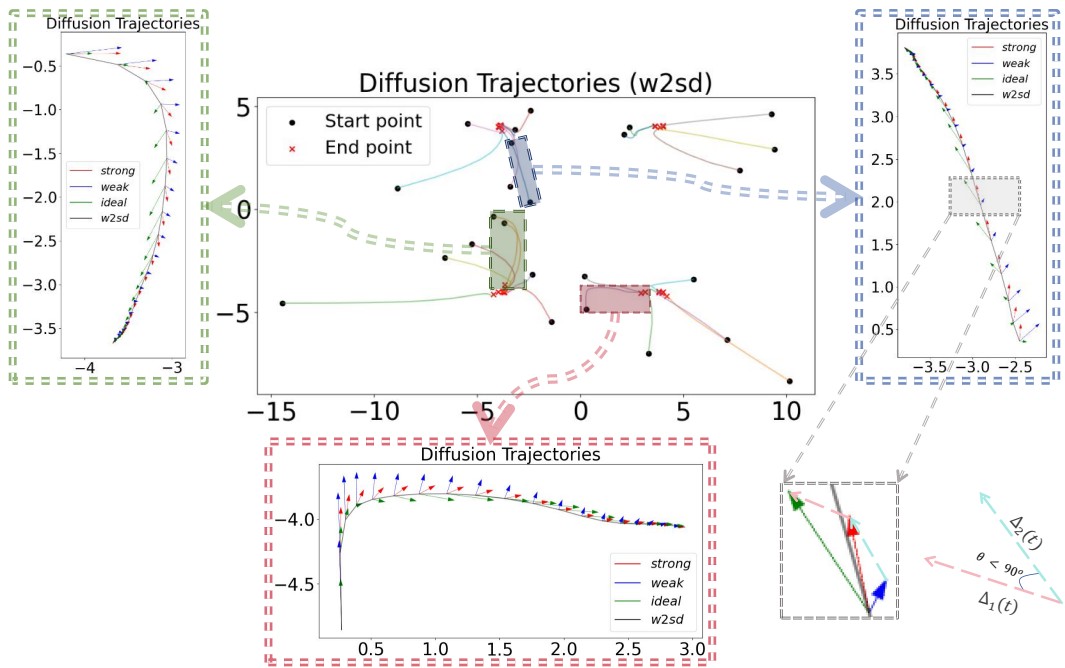

Figure 27: 2D denoising path visualization. The weak-to-strong gap can effectively bridge the strong-to-ideal gap.

Table 14: As the magnitude of the approximation error in inversion process increases, the gains from W2SD diminish. Model :xlMoreArtFullV1. Datasets: Pick-a-Pic. The type of W2SD: weight gap.

| Method | HPS v2 ↑ | AES ↑ | PickScore ↑ | MPS ↑ |
|---|---|---|---|---|
| SDXL | 29.8701 | 6.0939 | 21.6487 | - |
| xlMoreArtFullV1 | 32.8040 | 6.1176 | 22.3259 | 48.2224 |
| **W2SD (k=0)** | **33.5959** | 6.2252 | **22.3644** | **51.7770** |
| W2SD (k=0.005) | 32.9341 | **6.3228** | 22.3221 | 46.7130 |
| W2SD (k=0.010) | 19.5277 | 4.3949 | 17.9267 | 1.3440 |

this phenomenon occurs in both guidance gap and weight gap scenarios, indicating that minimizing the approximation error in inversion process is crucial.

### E.4 RELATIONSHIP WITH ITERATIVE SAMPLING

Re-Sampling (Lugmayr et al., 2022) stands as the pioneering iterative sampling algorithm in diffusion models. Numerous subsequent approaches including FreeDoM (Yu et al., 2023) and UGD (Bansal et al., 2023) have been developed based on this foundational work. We argue that these methods represent specialized instantiations of the W2SD framework.

We first confirm that Re-Sampling is generally a specific instance of W2SD, which can be interpreted within the framework of our theory in Theorem 1. In Figure 28, we note that Re-Sampling performs better when the randomly sampled Gaussian noise in Re-Sampling is similar to the perturbation vector introduced by the W2SD reflection mechanism (e.g., cosine similarity $> 0$). Additionally, we propose an advanced version of Re-Sampling (see Algorithm 4), which evaluates whether the similarity score, **sim_score**$(\epsilon_{\text{w2s}}, \epsilon)$, exceeds 0 to determine whether the Gaussian noise $\epsilon$ should be accepted for the ReNoise operation.

Table 15: By selecting Gaussian noise during the Re-Sampling process, the advanced Algorithm 4 achieves superior performance in the sampling process, demonstrating that Re-Sampling is a specific instance of W2SD. Model: SDXL. Datasets: Pick-a-Pic. The type of W2SD: guidance gap.

| Method | HPS v2 ↑ | AES ↑ | PickScore ↑ | MPS ↑ |
|---|---|---|---|---|
| SDXL | 29.8701 | 6.0939 | 21.6487 | - |
| Re-Sampling (sim score<0) | 30.2797 | 6.0744 | 21.6894 | 48.4793 |
| Re-Sampling (sim score>0) | 30.7844 | 6.0555 | **21.8620** | 51.5210 |
| **W2SD** | **31.2020** | **6.0970** | 21.7980 | **56.0608** |

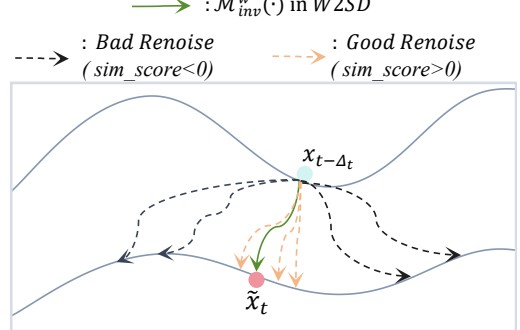

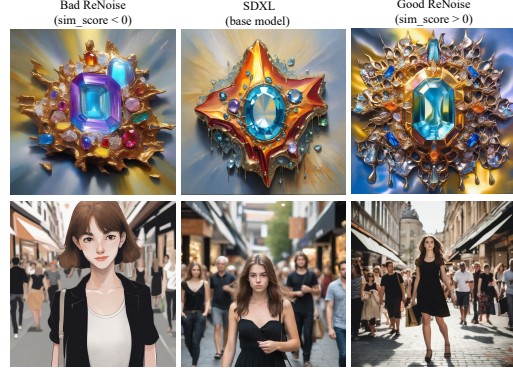

Figure 28: Re-Sampling, which can be considered a specific instance of W2SD, demonstrates improved performance when the randomly sampled Gaussian noise aligns closely with the perturbation vector introduced by the W2SD reflection mechanism (e.g., cosine similarity > 0)

Figure 29: Qualitative results of advanced Re-Sampling demonstrate that the improvements effects vary depending on the strategy used to select Gaussian noise for Re-Sampling. It can be considered a specific instance of W2SD.

We validate the effectiveness of Algorithm 4 in Table 15. When consistently selecting favorable random noise (i.e., $\epsilon_{\text{w2s}}, \epsilon) > 0$), advanced Re-Sampling demonstrates improved performance. Conversely, when consistently selecting unfavorable random noise (i.e., **sim_score**$(\epsilon_{\text{w2s}}, \epsilon) < 0$), the performance of Re-Sampling deteriorates. We also present the visualization results in Figure 29, further demonstrating that Re-Sampling can be incorporated into the W2SD framework, validating the correctness of our theory in Theorem 1.

Similarly, many subsequent research, such as FreeDoM (Yu et al., 2023), UGD (Bansal et al., 2023), MPGD (He et al., 2023), and TFG (Ye et al., 2024), follow the same approach as Re-Sampling by utilizing random Gaussian noise for iterative optimization. Therefore, these inference enhancement methods can be regarded as specific instances of W2SD. The primary distinction among these methods lies in the specific strong model $\mathcal{M}^s$ they employ. For instance, TFG utilizes a more refined parameter search mechanism, resulting in a strong model that exhibits greater robustness and performance compared to algorithms such as FreeDoM and UGD. As a consequence, under the condition of the same weak model (i.e., the addition of random Gaussian noise), TFG demonstrates significantly enhanced performance. However, these studies collectively overlook a fundamental aspect: the weak-to-strong gap constitutes the core principle that fundamentally drives the efficacy of such algorithms.

## E.5 RELATIONSHIP WITH AUTO-GUIDANCE

We note that in the weak-to-strong framework, Auto-guidance (Karras et al., 2024) employs a pre-trained diffusion model along with a corrupted version of it (typically achieved by adding perturbations or reducing training iterations through training from scratch). It directly enhances

**Algorithm 3** Vanilla Re-Sampling

**Input:** Strong Model $\mathcal{M}^{\mathrm{s}}$, Total Inference Steps: $T$, Optimization Steps: $\lambda$
**Output:** Clean Data $x_0$
Sample Gaussian noise $x_T$
**for** $t = T$ **to** 1 **do**
    **if** $t > T - \lambda$ **then**
        $x_{t-1} = \mathcal{M}^{\mathrm{s}}(x_t, t)$
        Initialize $\epsilon \sim \mathcal{N}(0, 1)$
        $x_t^{\mathrm{Re}} = \mathrm{Add\_Noise}(x_t, \epsilon, t)$
    **end if**
    $x_{t-1} = \mathcal{M}^{\mathrm{s}}(x_t^{\mathrm{Re}}, t)$
**end for**

**Algorithm 4** Advanced Re-Sampling

**Input:** Strong Model $\mathcal{M}^{\mathrm{s}}$, Weak Model $\mathcal{M}^{\mathrm{w}}$, Total Inference Steps: $T$, Optimization Steps: $\lambda$
**Output:** Clean Data $x_0$
Sample Gaussian noise $x_T$
**for** $t = T$ **to** 1 **do**
    **if** $t > T - \lambda$ **then**
        $x_{t-1} = \mathcal{M}^{\mathrm{s}}(x_t, t)$
        $\tilde{x}_t = \mathcal{M}^{\mathrm{w}}_{\mathrm{inv}}(x_{t-1}, t)$
        $\epsilon_{\mathrm{w2s}} = \tilde{x}_t - x_t$
        Calculate $\epsilon_{\mathrm{w2s}}$ based on Equation (5)
        #Select Optimal ReNoise
        Initialize $\epsilon \sim \mathcal{N}(0, 1)$
        **while** similarity_score $(\epsilon_{\mathrm{w2s}}, \epsilon) < 0$ **do**
            Initialize $\epsilon \sim \mathcal{N}(0, 1)$
        **end while**
        $x_t^{\mathrm{Re}} = \mathrm{Add\_Noise}(x_t, \epsilon, t)$
    **end if**
    $x_{t-1} = \mathcal{M}^{\mathrm{s}}(x_t^{\mathrm{Re}}, t)$
**end for**

performance by interpolating in the latent space. And here we clarify the contributions of W2SD in relation to it.

**Different mechanisms**   W2SD employs an reflection mechanism, while Auto-guidance utilizes an interpolation-based method. For comparison with W2SD, we set $w=1$ in Auto-guidance. When using Strong Model (human preference model) vs Weak Model (SDXL), in Table 16, we show that W2SD achieves notably higher scores on human preference metrics including PickScore and HPS v2. However, in this setting, direct interpolation in Auto-guidance leads to performance degradation in certain metrics, manifesting as oversaturation and artifacts. Actually, the auto-guidance mechanism similarly utilizes the following operations as

$$x_t \rightarrow x_{t-1}^{good} \tag{36}$$

$$x_t \rightarrow x_{t-1}^{bad} \tag{37}$$

$$x_{t-1}^{new} = x_{t-1}^{good} + w \cdot (x_{t-1}^{good} - x_{t-1}^{bad}). \tag{38}$$

When $w$ is too large, applying Equation (38) roughly refines the distribution of the latent variables, leading to an unnatural shift in the data distribution (Sadat et al., 2024; Lou & Ermon, 2023). When $w$ is too small, it fails to produce sufficient refinement.

Table 16: Performance comparison between Auto-guidance's interpolation and W2SD's reflection mechanism in latent variable refinement.

| Method | HPS v2 ↑ | AES ↑ | PickScore ↑ |
|---|---|---|---|
| SDXL | 29.8701 | 6.0939 | 21.6487 |
| xlMoreArtFullV1 | 32.8040 | 6.1176 | 22.3259 |
| Auto-guidance | 32.1650 | 6.1187 | 22.0177 |
| **W2SD** | **33.5959** | **6.2252** | **22.3644** |

In W2SD's process, $x_t \rightarrow x_{t-1} \rightarrow x_t$, the refinement operation (marked in red) is implicitly performed by score network's internal transformation which avoids common artifacts like distortion and over-saturation, please see Figure 30.

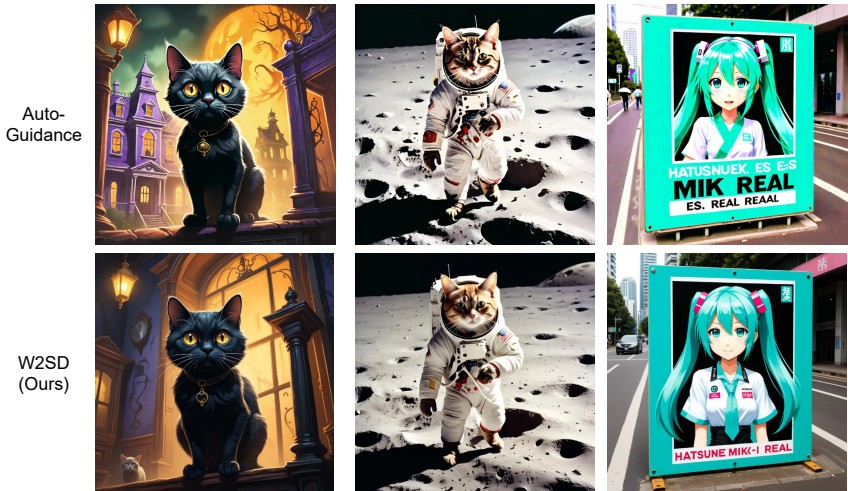

Figure 30: Compared with Auto-guidance, W2SD achieves superior performance with enhanced robustness, effectively addressing critical issues such as oversaturation and optimization failure.

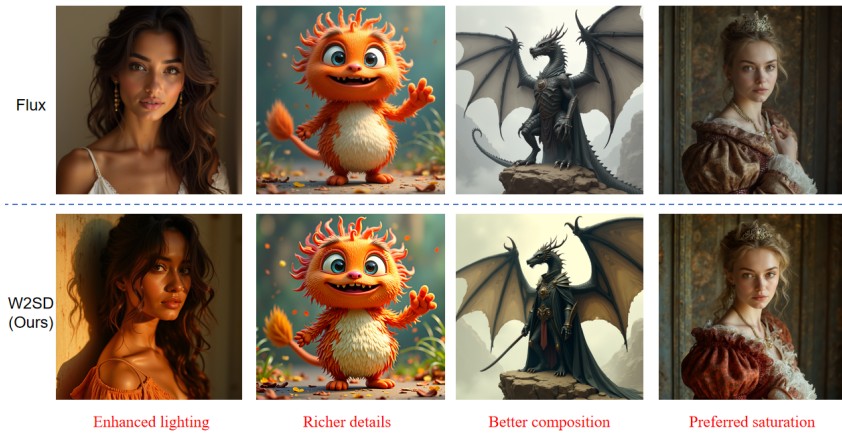

Figure 31: Visual Case of W2SD (Flux vs SDXL). W2SD can support cross-architecture operations (e.g., DiT and UNet).

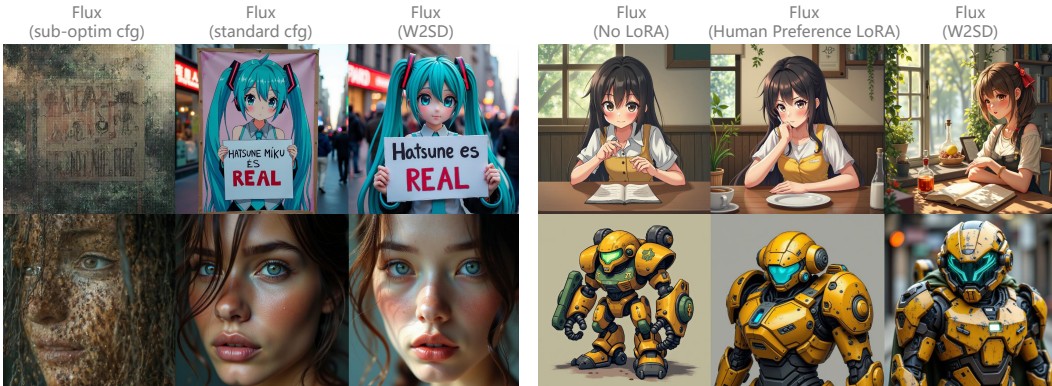

The results of W2SD Guidance Gap based on FLUX.1-dev          The results of W2SD LoRA Gap based on FLUX.1-dev

Figure 32: Flux model results: W2SD bridges the performance gap between weak and strong variants.

**Varying Applicability Ranges**   On the other hand, we note W2SD generalizes the concept of weak/strong model pairs—where the "weak" model is not limited to underperforming variants created through reduced capacity or training strategies (e.g., data corruptions or degradations as in Auto-guidance). We propose that gaps in semantic interpretation of prompts or sampling methodologies (e.g., MoE routers, ControlNet adaptations) can equally constitute valid weak-strong pairings. This expanded paradigm demonstrates significantly greater practical utility, as it accommodates real-world deployment scenarios where model capabilities vary along multiple axes.

**Cross-architecture Support**   Compared to Auto-guidance's direct interpolation approach, we demonstrate that W2SD's reflection mechanism enables cross-architecture operations in pixel space (mapping $x_t$ to pixel space ($x_0$), applying W2SD, and projecting back to latent space).

We select **Flux** as the strong model (latent shape: [1,16,128,128]) and **SDXL** as the weak model (latent shape: [1,4,128,128]). Despite the differences in latent shapes between Flux and SDXL, we note that W2SD can support cross-architecture operations in the pixel space (mapping $x_t$ to pixel space $x_0$), applying W2SD, and projecting back to latent space). In this configuration, we show that W2SD could improves image quality, with Figure 31, demonstrating superior lighting, detail preservation, better composition and preferred saturation. The image quality winning rate of W2SD is up to 74%. This suggests W2SD exhibits cross-architecture/data generality.

### E.6   EXPERIMENTS ON FLUX

In this section, we present supplementary experimental results of W2SD using Flux as the base model. Specifically, Table 17 and Table 18 report the results of two variants of W2SD, namely the *guidance gap* and *lora gap*. For the guidance gap variant, the weak model is set to CFG=1.0 and the strong model to CFG=2.0. For the LoRA gap variant, the weak model is the base model without any LoRA modules, while the strong model incorporates LoRA modules with a scale of 0.3. The total number of denoising steps $T$ is 28, with the reflection window $\lambda$ set to 10. Furthermore, we provide qualitative comparisons of the generated samples in Figure 32.

Table 17: Quantitative results of W2SD based on FLUX.1-dev. Datasets: Pick-a-Pic.

| Method | HPS v2 ↑ | AES ↑ | PickScore ↑ |
|---|---|---|---|
| Flux (sub-optimal cfg) | 23.2291 | 6.0081 | 20.5669 |
| Flux | 29.9532 | 6.3668 | 22.1578 |
| **W2SD (based on guidance gap)** | **31.1962** | **6.3677** | **22.2464** |
| Flux (+ LoRA) | 30.3402 | **6.3987** | 22.2199 |
| **W2SD (based on lora gap)** | **31.7411** | 6.3580 | **22.3050** |

Table 18: Quantitative results of W2SD based on FLUX.1-dev. Datasets: DrawBench.

| Method | HPS v2 ↑ | AES ↑ | PickScore ↑ |
|---|---|---|---|
| Flux (sub-optimal cfg) | 22.5098 | 5.5170 | 21.4877 |
| Flux | 29.2550 | 5.8035 | 22.8866 |
| **W2SD (based on guidance gap)** | **30.6314** | **5.9030** | **23.0236** |
| Flux (+ LoRA) | 29.5438 | 5.8088 | 22.8680 |
| **W2SD (based on lora gap)** | **30.9390** | **5.9238** | **22.9202** |

