# OpenReview forum: "Weak-to-Strong Diffusion with Reflection"
_ICLR.cc/2026/Conference — ICLR 2026 Poster_

### Official Review · Reviewer_nwAH · 2025-10-31

**Soundness:** 3
**Presentation:** 4
**Contribution:** 3
**Rating:** 6
**Confidence:** 4

**Summary:**

- This paper introduces Weak-to-Strong Diffusion (W2SD), a novel, training-free framework designed to enhance the performance of diffusion models during inference. The core problem it addresses is the "modeling gap": the discrepancy between the data distribution learned by a diffusion model and the true, ideal data distribution.
- Essentially,  instead of trying to directly estimate this unattainable "strong-to-ideal" gap, W2SD leverages a more accessible proxy: the "weak-to-strong" gap. This is the difference in behavior between a high-performing "strong" model (e.g., a fine-tuned model like Juggernaut-XL) and a lower-performing "weak" model (e.g., a base model like SDXL).
- The framework operates through a reflection mechanism. During sampling, it alternates between:
  - A denoising step using the strong model (to move towards a high-quality output).
  - An inversion step using the weak model (to add back a controlled amount of noise, effectively "reflecting" the latent variable).
- A key contribution is the broad definition of what constitutes a "weak" and "strong" model. The paper demonstrates three types of gaps:
  - **Weight Gap:** Different model weights (e.g., SD1.5 vs. DreamShaper, base model vs. a LoRA-adapted model, strong vs. weak experts in a Mixture-of-Experts model).
  - **Condition Gap:** The same model under different conditions (e.g., high vs. low classifier-free guidance scale, detailed vs. simple text prompts).
  - **Sampling Pipeline Gap:** Different inference pipelines (e.g., ControlNet/IP-Adapter vs. standard DDIM sampling).

  As shown in the paper, when combined, they usually yield better results.

- Extensive experiments across image and video generation tasks show that W2SD consistently improves performance on metrics like human preference (HPS v2, PickScore), aesthetics (AES), and fidelity (FID), often outperforming related methods like Auto-Guidance and Z-Sampling. Crucially, the performance gains are shown to outweigh the additional computational cost, making W2SD efficient under equal time constraints.

**Strengths:**

- The paper is well written and easy to follow.
- **High Flexibility and Generality:** W2SD is not a single technique but a meta-framework. Its ability to incorporate various types of model pairs (weights, conditions, pipelines) makes it widely applicable across many use cases without retraining.
- **SOTA Empirical Results:** The paper provides comprehensive quantitative and qualitative evidence across multiple benchmarks (Pick-a-Pic, DrawBench, ImageNet), architectures (UNet, DiT), and tasks (image, video generation), demonstrating consistent and significant improvements.
- **Training-Free:** As an inference-time method, W2SD can be readily applied to existing models and pipelines without the need for costly additional training.
- **Diverse evaluation:** The method is tested on many datasets, different diffusion models, and various types of diffusion use-cases.
- **Ablation Studies:** The method is properly analyzed on both synthetic and real data cases. Each part of the method is thoroughly interpreted, highlighting its effectiveness. Additionally, a toy test is shown in the paper (Figure 6), demonstrating the benefit of the method, where failure cases of both the strong and the weak models were exploited to produce pleasing-looking images of cars.

**Weaknesses:**

- The contribution of the result from Theorem 1 is minimal: It states a very naive result, without adding any contribution to the paper or supporting the method's effectiveness in any way.
- The method assumes that the score function is very smooth w.r.t. t, specifically the approximation in (4). This might be a limiting factor of the method in accelerated diffusion setups, where the jump between two consecutive steps is more significant.
- The method is not supported by theoretical justification.
- Performance Degradation: If the weak-to-strong gap is not a good proxy for the strong-to-ideal gap (or is even negatively aligned), the method can lead to worse results than the strong model alone.
- Limited Exploration of Limitations: Some practical limitations are mentioned only briefly in the appendix. A deeper discussion of failure modes or scenarios where W2SD is not applicable would be beneficial.

**Questions:**

- How does this method perform when using a smaller number of diffusion steps? Does the assumption of the smoothness of the score degrade the generation performance?
- Is it possible to theoretically justify this improvement?

---

> ### Author Response · Authors · 2025-11-23
> **Response to Reviewer nwAH - Part 1**
>
> We sincerely thank the reviewer for the constructive and insightful comments. We appreciate your recognition of our work. Below, we provide responses to your main concerns.
>
> ---
>
> **W1**. The contribution of the result from Theorem 1 is minima.
>
> We thank the reviewer for their feedback on Theorem 1 and acknowledge that its derivation is computationally straightforward. We would like to emphasize that its purpose is to serve as a conceptual bridge, offering an intuitive interpretation that the W2S mechanism guides the latent variable toward regions of higher likelihood under the strong model, while steering it away from the weak model’s distribution. This formulation was essential in establishing a coherent narrative that connects Theorem 1 to the supporting visual evidence presented in our work.
>
> Following the reviewer’s suggestion, we have further strengthened our theoretical analysis by introducing **Theorem 2**, which analytically characterizes the conditions under which Δ₁ can serve as a reliable proxy for Δ₂. To complement this theoretical contribution, we have also added **Fig. 3**, which provides experimental validation of these findings. Additional related discussion can be found in our response to comment W.3.
>
> ---
>
> **W2**.How does this method perform when using a smaller number of diffusion steps?
>
> In response to the reviewer's suggestion, we conducted evaluations using DreamShaper-XL-Turbo (with 4-step generation) and DPM-Solver++ as the base model. The results demonstrate that W2SD with guidance gap consistently outperforms standard sampling across all evaluated metrics on both the Pick-a-Pic and DrawBench benchmarks.
>
> | Benchmark: Pick-a-Pic | HPS v2 | AES | PickScore | ImageReward |
> |---------|---------|---------|---------|---------|
> | DreamShaper-xl-v2-turbo |  30.04   |  5.93   |  21.59   |  66.18   |
> | W2SD |   32.38     |   6.15     |   22.11    |  90.87   |
>
> | Benchmark: DrawBench | HPS v2 | AES | PickScore | ImageReward |
> |---------|---------|---------|---------|---------|
> | DreamShaper-xl-v2-turbo |   26.85   |  5.28    |   21.77  |   40.22     |
> | W2SD |    29.90    |    5.64     |    22.35     |   73.51   |
>
> We further performed a trajectory visualization analysis for the 1D case in **Fig. 21**. While the 5-step W2S trajectory does exhibit local irregularity (Fig. 21, top-left), as rightly noted by the reviewer, its macro-level alignment with the weak–strong directional consensus ensures effective overall guidance. This behavior confirms W2SD’s ability to steer the sampling process toward desirable regions even under limited-step conditions.
>
> ---
>
> **W3**. Is there more theoretical justification for the method?
>
> Yes. In response to the reviewer’s comment, we have extended our theoretical analysis regarding the conditions under which the guidance gap can serve as a reliable proxy.
>
> - We established an analytical relationship between the weak-to-strong gap and the strong-to-ideal gap in **Theorem 2**. The criterion for Δ₁ to act as a valid proxy for Δ₂ is formally given by Eq. (8) and Eq. (9). The validity of this theorem is corroborated both analytically and experimentally, as supported by the results presented in **Fig. 3**. Specifically, given exact knowledge of any two of the three models—ideal, strong, and weak—the third can be derived analytically.
>
> - In **Appendix E.2**, we further compute the cosine similarity between Δ₁ and Δ₂ throughout the denoising process on CIFAR-10. As shown in **Fig. 26**, the similarity remains consistently positive across all timesteps, reinforcing the practical relevance of our theoretical findings.
>
> The inclusion of these new analyses, along with extensive experimental validation, strengthens the foundation of our method and confirms the robustness of its underlying assumptions. We thank the reviewer for their insightful comment, which has helped us improve the rigor and completeness of this work.
>
> ---

---

> > ### Author Response · Authors · 2025-11-23
> > **Response to Reviewer nwAH - Part 2**
> >
> > ---
> >
> > **W4**. Could the weak-to-strong gap lead to model degradation if it's a bad proxy?
> >
> > Yes. The newly added visualization case in **Fig.3** demonstrates that when the weak-to-strong gap fails to meet the conditions specified in Theorem 2, it cannot serve as a reliable proxy. Furthermore, based on real-world data, we have empirically identified two scenarios where W2SD is not applicable:
> >
> > - Model conflict (see **Fig.22**): Occurs when the weak and strong models possess conflicting or mutually exclusive capabilities, leading to contradictory guidance signals that hinder effective optimization.
> >
> > - Model over-similarity (see **Fig.23**): Occurs when the capability gap between models is too small, resulting in an insufficiently weak guidance signal that fails to provide meaningful performance improvements.
> >
> > It's worth noting that in our main experiments, all evaluated model pairs maintain natural weak and strong distinctions (e.g., guidance/pipeline gaps). These configurations are both intuitive to identify and inherently resilient to such failure scenarios.
> >
> > ---
> >
> > **W5**. Are there any further discussions regarding the failure modes or scenarios?
> >
> > Yes. Following the reviewer's suggestion, we have conducted a systematic investigation into the failure modes of W2S in extreme scenarios, supported by both theoretical analysis and empirical experiments. A detailed discussion is provided in our response to comment W4.
> >
> > ---
> >
> > **Q1**.  How does this method perform when using a smaller number of diffusion steps?
> >
> > W2SD maintains robust performance even in few-step generation settings (as few as 4 steps). Comprehensive results and analyses are provided in our response to comment W2.
> >
> > ---
> >
> > **Q2**. Is it possible to theoretically justify this improvement?
> >
> > Yes. We have provided both enhanced theoretical analysis and supporting empirical results to substantiate this improvement. Please see our response to comment W3 for a complete discussion.
> >
> > ---
> >
> > Finally, we sincerely thank the reviewer's feedback. It inspires us to further improve our work and clarification for more readers.  We are more than happy to discuss if the reviewer has further questions or comments.

---

> > > ### Comment · Reviewer_nwAH · 2025-11-24
> > >
> > > I would like to thank the authors for their detailed responses and for addressing most of my concerns.
> > >
> > > Regarding **Theorem 2**:
> > > - As far as I can tell, the purpose of this result is to provide a theoretical proxy for deciding when to use the weak model based on its performance relative to the strong model, which indeed strengthens the overall approach.
> > > - However, I still do not see a theoretical justification for why this method is preferable to using only the strong model. In particular, the contribution would be significantly stronger if you could bound the total error of the W2SD method and demonstrate that it is smaller than the error of the strong model (and, effectively, the weak model).
> > >
> > >
> > > Finally, I kindly ask that the smiling-face emojis be removed from the figures to maintain a formal presentation.

---

> ### Author Response · Authors · 2025-11-25
> **Response to "Total Error Bound and Comparison with Strong Model"**
>
> This is an insightful comment. We agree that demonstrating the theoretical advantage of W2SD over the standalone Strong Model is crucial.
>
> To demonstrate that W2SD yields a smaller total error than the strong model, we analyze the Fisher Divergence $$ between the model distribution and the ground truth. In diffusion models, the Fisher Divergence acts as a proxy for the KL Divergence and serves as the canonical "Total Error" metric. To meet the length limit, Theorem 3 and its proof have been moved to **Appendix A.3**. The key steps are available on OpenReview.
>
> ---
>
> **Theorem 3 (Global Error Reduction via Fisher Divergence).** *Let $p^{\mathrm{gt}}(x)$ denote the ground truth data distribution. Let $\mathcal{J}(p^{\mathrm{gt}} \| p^{\theta}) = \frac{1}{2} \mathbb{E}_{x \sim p^{\mathrm{gt}}} [ \| \nabla_x \log p^{\mathrm{gt}}(x) - \nabla_x \log p^{\theta}(x) \|^2 ]$ be the Fisher Divergence measuring the total score matching error.*
>
> *Consider the refined score estimate of W2SD defined as $s_{\mathrm{w2sd}}(x) = \nabla_x \log p^{\mathrm{s}}(x) + \Delta_1(x)$. Under the approximation condition established in Theorem 2, where the point-wise estimation error is bounded by a factor $\epsilon \in [0, 1)$ such that $\| \Delta_1(x) - \Delta_2(x) \| \leq \epsilon \| \Delta_2(x) \|$, the total error of W2SD is strictly bounded by the total error of the strong model:*
>
> $$\mathcal{J}(p^{\mathrm{gt}} \| p^{\mathrm{w2sd}}) \leq \epsilon^2 \mathcal{J}(p^{\mathrm{gt}} \| p^{\mathrm{s}}) < \mathcal{J}(p^{\mathrm{gt}} \| p^{\mathrm{s}}).$$
>
> *This implies that W2SD theoretically achieves a globally superior generation quality compared to the standalone strong model.*
>
> ---
>
> **Key Proof.** The Fisher Divergence for the strong model $\mathcal{M}^{\mathrm{s}}$ is given by the expectation of the squared norm of the strong-to-ideal gap $\Delta_2(x)$:
>
> $$\mathcal{J}(p^{\mathrm{gt}} \| p^{\mathrm{s}}) = \frac{1}{2} \int p^{\mathrm{gt}}(x) \| \Delta_2(x) \|^2 dx.$$
>
> For the W2SD framework, the refined score is $s_{\mathrm{w2sd}}(x) = \nabla_x \log p^{\mathrm{s}}(x) + \Delta_1(x)$, and the corresponding error is $\| \Delta_2(x) - \Delta_1(x) \|^2$. Based on the premise derived from Theorem 2 (relative bias consistency), we have the point-wise bound $\| \Delta_2(x) - \Delta_1(x) \| \leq \epsilon \| \Delta_2(x) \|$. Substituting this inequality into the integral:
>
> $$\begin{aligned}
> \mathcal{J}(p^{\mathrm{gt}} \| p^{\mathrm{w2sd}}) &= \frac{1}{2} \int p^{\mathrm{gt}}(x) \| \Delta_2(x) - \Delta_1(x) \|^2 dx \\
> \leq \frac{1}{2} \int p^{\mathrm{gt}}(x) \left( \epsilon \| \Delta_2(x) \| \right)^2 dx = \epsilon^2 \mathcal{J}(p^{\mathrm{gt}} \| p^{\mathrm{s}}).
> \end{aligned}$$
>
> Since $0 \leq \epsilon < 1$ holds for well-aligned weak-to-strong pairs, it follows that $\mathcal{J}(p^{\mathrm{gt}} \| p^{\mathrm{w2sd}}) < \mathcal{J}(p^{\mathrm{gt}} \| p^{\mathrm{s}})$.
>
> ---
>
> Finally, in response to the reviewer's constructive input, we have omitted the smiley face from Fig. 3. We sincerely appreciate this suggestion, which has helped to better underscore the substantive contributions of our work, and we hope that all of our responses have now fully addressed your concerns.

---

> > ### Comment · Reviewer_nwAH · 2025-11-27
> >
> > I would like to thank the authors for their detailed responses and for addressing my concerns. As all of my concerns have now been resolved, I am updating my rating accordingly.
> >
> > As a final recommendation, I suggest moving Theorem 3 from the appendix into the main paper to improve clarity and visibility.

---

> > > ### Author Response · Authors · 2025-11-28
> > > **Authors' Response**
> > >
> > > We are especially grateful for your positive evaluation and the subsequent elevation in your score. In accordance with your final suggestion, Theorem 3 will be moved into the main body of the paper in the final version to enhance its visibility and improve the logical flow.

---

### Official Review · Reviewer_Xkjj · 2025-10-31

**Soundness:** 4
**Presentation:** 4
**Contribution:** 3
**Rating:** 8
**Confidence:** 5

**Summary:**

This paper introduces a new approach to minimize the gap between the learned probability distribution and the actual probability distribution(intractable) of data. This uses the divergence of distribution of a weak model and a strong model as the proxy for divergence of real and learned distributions.

**Strengths:**

1. Simple and general idea with practical reach. Using the difference between a weak and a strong model as proxy for the actual difference between real and learned distribution seems to be fair given the assumptions in the paper.
2. The W2SD algorithm is backed by various experiments, showcasing that it can be practically used with pretrained models.

**Weaknesses:**

1. The central claim is that the weak-to-strong gap $\Delta_1$ is a useful proxy for the intractable strong-to-ideal gap $\Delta_2$. The paper provides cosine-similarity evidence on CIFAR, but this may not generalize: for some kinds of differences (e.g., when $M_w$ is biased towards different modes than the ideal), $\Delta_1$ might be orthogonal or adversarial to $\Delta_2$. The presented sensitivity analyses (gap sign and magnitude) indicate failure modes, but there is not a principled rule for selecting pairs to guarantee positive gains.
2. Even though the method shows gains in various quantitative scores, the increase in computation is not described in paper and is an important factor when comparing with techniques such as Z-sampling, AutoGuidance, CFG, etc.

**Questions:**

1. What is the increase in computation for this method as compared to the SOTA methods such as Z-sampling, AutoGuidance etc?
2. What is the relation between the values of $\mathcal{T}$ and $\lambda$? How are these decided based on the models? A discussion on this would be helpful as to how these parameters handles the proxy.
3. Since LoRA like techniques are used for personalization i.e., to bias a model towards a specific distribution, what is the intuition behind considering the model strong? Strong in what terms?

---

> ### Author Response · Authors · 2025-11-23
> **Response to Reviewer Xkjj**
>
> We appreciate the reviewer for the thorough feedback, and are delighted that our work has received your recognition. Below we address each of the reviewer's concerns.
>
> ---
>
> **W1**. The limitations of using the weak-to-strong gap as a proxy for the strong-to-ideal gap.
>
> We thank the reviewer for their insightful comments regarding our cosine similarity experiments. To further support the use of Δ₁ as a proxy measure, we have introduced **Theorem 2** in the revised manuscript, which establishes an analytical relationship between Δ₁ and Δ₂ and provides a theoretical error bound. This theoretical foundation is empirically corroborated by the experiments in **Fig. 3**, where we demonstrate that Δ₁ serves as an effective approximation of Δ₂ under commonly encountered conditions.
>
> We also acknowledge that discrepancies between Δ₁ and Δ₂ may occur in extreme cases. To address this, we have identified and analyzed two specific failure modes—model conflict and model over-similarity—in **Figs. 22 & 23**, supported by validation on real images. The model pairs selected for our main experiments reflect typical scenarios (such as those involving the guidance gap), thereby situating our study within a regime where Δ₁ remains a reliable proxy. This ensures the robustness and validity of our findings.
>
> ---
>
> **W2**. The corresponding increase in computational cost
>
> W2SD combines efficiency with strong performance: it matches the speed of Z-Sampling while surpassing standard sampling (**Fig. 14**), and its temporal scaling brings large gains from little cost (**Figs. 19-20**). Auto-guidance reduces computational cost by one-third; notably, however, this gain is accompanied by significant quality compromises, as analyzed in **Appendix E.5**.
>
> ---
>
> **Q1**.What is the increase in computation for this method as compared to other methods?
>
> The computational overhead of W2SD is  explained in W2.
>
> ---
>
> **Q2**.What is the relation between the values of $\lambda$ and T?
>
> In our method, T denotes the total number of denoising steps in the diffusion inference process, while λ controls how many of these steps are applied to the reflection operation.
>
> As an inference-time scaling algorithm, both empirical observations and intuitive reasoning suggest that W2SD generally achieves better performance with higher values of λ. We explore the time–performance trade-off curve of W2SD in detail in **Figs.19-20**.
>
> ---
>
> **Q3**. what makes one model "stronger" than another, and in what sense?
>
> In our framework, the notion of a "strong" model is inherently task-dependent. It is defined by a model's relative advantage with respect to a specific objective. For instance, if the goal is to generate aesthetically pleasing rather than photorealistic images, the model specializing in aesthetics is considered the strong one, while the model focused on realism is treated as the weak counterpart.
>
> Building on this conceptual foundation, we propose a simple yet effective enhancement to W2SD in **Algorithm 2**. By allowing users to define a custom reward metric, this extended version can automatically determine which model serves as the strong counterpart and which as the weak during the denoising process.
>
> ---
>
> Finally, we sincerely thank the reviewers Xkjj for their constructive comments and recognition of our work. We hope these additional explanations resolve the reviewers' concerns.

---

> > ### Comment · Reviewer_Xkjj · 2025-11-24
> >
> > I would like to thank the authors for addressing my questions.
> >
> > However, in **W1**, the point I intended to make is that while this method generalizes well for similar probability distributions, as assumed in Theorem 2(GMMs), it may not work for dissimilar cases. I do not have any specific concerns regarding this, but I would appreciate it if the authors could include a brief explanation of this limitation in the main paper rather than only in the supplementary section.

---

> > > ### Author Response · Authors · 2025-11-25
> > > **Response to "Explanation of this Limitation in the main paper"**
> > >
> > > Thank you for your valuable suggestions. We have reorganized the manuscript accordingly. In the newly added comprehensive discussion section (Lines 206-211), we systematically elaborate on the two failure modes of W2SD and rigorously align them with the theoretical conclusions established in Theorem 2.

---

> > > > ### Comment · Reviewer_Xkjj · 2025-11-25
> > > >
> > > > Thank you for the addition. I already gave a high score to this paper and will maintain it. All the best!

---

> > > > > ### Author Response · Authors · 2025-11-25
> > > > > **Wishing you all the best as well**
> > > > >
> > > > > Thank you so much for your kind note and for the positive review! We really appreciate it.
> > > > >
> > > > > Wishing you all the best as well

---

### Official Review · Reviewer_99Dw · 2025-10-31

**Soundness:** 3
**Presentation:** 2
**Contribution:** 3
**Rating:** 4
**Confidence:** 4

**Summary:**

This paper introduces Weak-to-strong diffusion models (W2SD), an inference‑time framework that uses a reflection operator combining a strong diffusion model (M_s) and a weak model (M_w) to push samples along the estimated weak‑to‑strong gradient gap.

The proposed framework is flexible in choosing weak/strong pairs, including weight gaps, condition gaps, MoE routing gaps, and pipeline gaps. Experiments cover text‑to‑image, personalized LoRAs, and even video generation, showing consistent improvements and cumulative gains across gaps

**Strengths:**

- The unified W2SD is general and applicable to different diffusion models.
- Strong results covering different regions of interest.

**Weaknesses:**

Overall, the presentation of this paper is a problem. There are many figures and tables in both the main paper and the supplementary. However, the authors failed to organize them well in the text, making the paper slightly hard to read. A few concrete comments include:
- Figure 2 is hard to understand. No further explanations are associated with it.
- Algorithm 1 did not show anything specific about the W2SD algorithm, but a widely used procedure for inference-based optimizations.
- Eq 5 and the sentence below it are inconsistent, i.e., no \delta_1(t) at all.
- Also, Eq 5 cannot be called a "Theorem". Even with Section A in the Appendix, altogether they are a definition but not a strict proof.
- l135-137: It is unfair to treat Z-Sampling (Bai et al., 2024) and other similar methods as instances of W2SD.

Other weaknesses:
- Comparison to stronger baselines. For most of the experiments, the authors compare the W2SD version with the corresponding baseline model. It is also better to include other baselines. For example, with W2SD, can we tweak SDXL to Flux?
- Mathematical support for the correctness of W2SD is missing.

**Questions:**

- How does W2SD compare to Auto‑Guidance and Z‑Sampling **under matched compute** across diverse prompts (quantitatively)?
- Does repeated reflection risk **mode collapse** in long runs (i.e., is there guaranteed convergence?)?
- Theoretically, how to justify or prove the claim of bridging the strong-to-ideal gap with the weak-to-strong gap?
- How to precisely define a weak, strong, and idea model in the framework, for example, for specific scenarios, a common weak model can be "strong".

---

> ### Author Response · Authors · 2025-11-23
> **Response to Reviewer 99Dw - part 1**
>
> We are truly grateful for the time and thoughtful efforts the reviewer has devoted to evaluating our work and providing such constructive comments. Below, we provide detailed explanations to address your concerns.
>
> ---
>
> **W1**. The presentation of this paper
>
> We thank the reviewer for pointing out these issues with expression. We will address each point as follows in the final version:
>
> - **(a)**. Figure 2 is hard to understand.
>
>   We thank the reviewer for the valuable feedback on the figure. We will further improve the clarity of Fig. 2 in the  future version. Here, $\mathcal{M}^{s}$ and $\mathcal{M}^{w}$ denote the strong and weak models, respectively, with brown arrows representing the denoising operation and green arrows indicating the inversion operation.
>
> - **(b)**. The specifics of the Algorithm 1
>
>   We thank the reviewer for this point. Algorithm 1 provides a general framework that offers significant flexibility in defining strong and weak models. Building on this foundation, we have developed several algorithmic variants—including Auto-selection W2SD (**Algorithm 2**) and Re-Sampling+W2S (**Algorithm 4**)—to better demonstrate the scalability and distinctive advantages of our proposed approach.
>
> - **(c)**. Eq.5 and the sentence below it are inconsistent
>
>   We appreciate the reviewer's attention to detail. To improve readability, we have added further explanatory notes in L175–177 to help elucidate this relationship.
>
> - **(d)**. Equation 5 is a definition, not a theorem.
>
>   We fully understand the reviewer's concern and would like to clarify that Theorem 1 primarily serves as a conceptual bridge to explain the physical intuition behind W2SD. Following the reviewer's suggestion, we have further strengthened our theoretical analysis in subsequent (W3 and Q3) to provide more rigorous analytical support.
>
> - **(e)**. It is unfair to treat Z-Sampling and other similar methods as instances of W2SD.
>
>   We appreciate the reviewer's perspective on this comparison. To clarify, our intention is not to claim Z-Sampling as a strict instance of W2SD, but rather to position W2SD as a generalized framework that shares conceptual similarities with Z-Sampling in its use of model collaboration. We will revise the content to more precisely articulate this relationship and ensure a fair representation of Z-Sampling and related methods.
>
> ---
>
> **W2**. More baseline (e.g., W2S on Flux)
>
> We thank the reviewer for their valuable suggestion. Due to the comprehensive nature of W2SD as a general framework and the substantial experimental workload involved, we plan to report W2SD's experimental results on Flux within the next few days. We agree that this additional validation will further substantiate the method's effectiveness and general applicability.
>
> ---
>
> **W3**. Mathematical support for W2SD
>
> We thank the reviewer for their feedback on Theorem 1. While acknowledging its computational simplicity, we emphasize its role as a conceptual bridge: it provides physical intuition that the W2S mechanism guides the latent toward regions of higher likelihood under the strong model while steering it away from the weak model's distribution. This foundation was essential for constructing a coherent narrative connecting Theorem 1 to the supporting visual evidence in our work.
>
> We agree that expanded theoretical analysis strengthens our contribution and have updated the manuscript accordingly:
>
> - Applicability Conditions (**Theorem 2**): We added a theoretical analysis of W2SD's applicability validated by exact analytical solutions in **Fig. 3**. We demonstrate that determining any two models within the triad (ideal, strong, weak) analytically derives the third. This finding also grounds our summary of real-world failure modes (detailed in Q3).
>
> - Error Reduction Via Fisher Divergence (**Theorem 3**): We prove that W2SD achieves a lower Fisher Divergence than the standalone strong model. Under the conditions of Theorem 2, the total error satisfies $\mathcal{J}(p^{\mathrm{gt}} \| p^{\mathrm{w2sd}}) \leq \epsilon^2 \mathcal{J}(p^{\mathrm{gt}} \| p^{\mathrm{s}}) < \mathcal{J}(p^{\mathrm{gt}} \| p^{\mathrm{s}})$, theoretically guaranteeing improved generation quality.
>
> We hope these discussions further substantiate the theoretical contributions of W2SD.
>
> ---
>
> **Q1**. How do W2SD, Auto‑Guidance, and Z‑Sampling compare given equal compute?
>
> Both Z-Sampling and W2SD require the same inference time. Notably, within this fixed computational budget, W2SD can simultaneously leverage multiple model gaps—such as combining guidance and LoRA gaps—enabling it to achieve superior performance compared to Z-Sampling (**Tab.7**).
>
> While Auto-guidance uses two-thirds of Z-Sampling's inference time, it performs even worse than standard sampling on real-world data (**Tab.16**) and exhibits obvious over-saturation artifacts (**Fig. 30**). In contrast, W2SD effectively alleviates such over-saturation while maintaining competitive efficiency.
>
> ---

---

> ### Author Response · Authors · 2025-11-23
> **Response to Reviewer 99Dw - part 2**
>
> ---
>
> **Q2**. Does repeated reflection risk mode collapse in long runs ?
>
> Thanks reviewer for this question regarding "repeated reflection." We would like to clarify the interpretation from two perspectives:
>
> If the term refers to extending the reflection window, our experiments in **Figs. 19** and **Fig. 20** demonstrate that W2SD follows an inference time scaling law, where performance systematically improves with increased inference duration.
>
> If instead it refers to performing multiple rounds of W2S optimization at the same noise level (as implemented in our supplementary code), we note that prolonged iterations (e.g., three or more consecutive cycles) may lead to performance degradation. However, in most practical scenarios, a single reflection iteration already achieves satisfactory results.
>
> ---
>
> Q3. The claim of bridging the strong-to-ideal gap with the weak-to-strong gap
>
> We fully understand the reviewer's concern. In response, we have included in the revised manuscript a detailed analysis of the conditions under which the weak-to-strong gap can serve as a reliable proxy for the strong-to-ideal gap (see **Theorem 2**).
>
> And **Fig.3** utilizes the analytical solution of a toy distribution to visually validate the criteria from Theorem 2 for the validity of the Δ₁ proxy. Grounded in this analysis, we delineate two specific extreme scenarios on real image data where W2SD is not applicable: Model conflict (**Fig.22**) and Model over-similarity (**Fig.23**). It's worth noting that in our main experiments, all evaluated model pairs maintain natural weak and strong distinctions (e.g., guidance/pipeline gaps). These configurations are both intuitive to identify and inherently resilient to such failure scenarios.
>
> Furthermore, **Fig.26** shows that the cosine similarity between the weak-to-strong gap and the strong-to-ideal gap remains mostly positive during denoising (CIFAR-10), indicating general directional consistency between them.
>
> ---
>
> **Q4**. How to precisely define a weak, strong, and idea model in the framework
>
> In Line 99, we note that users can define weak-to-strong model pairs according to their specific needs. This implies the "weak" or "strong" concept is not limited to strict training data recovery (as shown in Sec 3.2's motivation experiments and discussed in [1]), but can be viewed as a relatively "stronger" model.
>
> | Method Type      | $\mathcal{M}^{1}$ | $\mathcal{M}^{2}$ | $\mathcal{M}^{1}$ ( $\mathcal{M}^{2}_{\text{inv}}$ ) | $\mathcal{M}^{2}$ ( $\mathcal{M}^{1}_{\text{inv}}$ )  | Auto-Selection (W2SD) |
> |-----------------|-------------|-------------|-----------|-----------|----------------------|
> | HPS v2 Score    | 30.16       | 26.66       | 27.16     | 30.29     | 30.89                |
>
> To further clarify, our method completely eliminates the need for manual strong/weak model selection. We propose a W2SD variant (**Algorithm 2**) where users only need to define an optimization objective (e.g., human preference) and provide a corresponding metric model (e.g., HPS v2). This approach leverages the reward model to automatically designate strong and weak models at each denoising step by evaluating intermediate outputs at noise level σ=0, enabling real-time assessment and dynamic model selection. We report our results in **Table 13** and **Fig. 24**.
>
> ---
>
> Finally, we sincerely thank the reviewer for their constructive feedback. We hope the additional experiments and explanations provided address the reviewer’s concerns, and we respectfully invite you to reconsider our submission.
>
> ---
>
> [1] Karras, Tero, et al. Guiding a diffusion model with a bad version of itself. Neruips 2024.

---

> ### Comment · Reviewer_99Dw · 2025-11-26
> **Post rebuttal response**
>
> I thank the authors for providing such a detailed response. Most of my concerns have been addressed. I would like to increase my score to 6. However, I will not champion the acceptance.

---

> > ### Author Response · Authors · 2025-11-28
> > **Gratitude for the Final Decision and Review**
> >
> > Thank you for your thoughtful review and for confirming that the revisions have addressed most of your concerns, resulting in the elevated score.
> >
> > Wishing you all the best.

---

### Official Review · Reviewer_DEkr · 2025-11-01

**Soundness:** 3
**Presentation:** 3
**Contribution:** 3
**Rating:** 6
**Confidence:** 4

**Summary:**

This paper introduces Weak-to-Strong Diffusion (W2SD), a novel, training-free framework designed to enhance the performance of pre-trained diffusion models at inference time. The core idea is to leverage the "weak-to-strong gap"—the difference in score estimates between a capable (strong) model and a less capable (weak) one—as a proxy for the unattainable "strong-to-ideal" gap. The authors propose a reflective sampling mechanism, which alternates between denoising with the strong model and inverting with the weak model, to steer the sampling trajectory towards the true data distribution. W2SD is presented as a general meta-framework, and its effectiveness is demonstrated across a wide variety of "weak-strong" pairings, including gaps in model weights, conditional inputs (e.g., CFG scale, prompt quality), and sampling pipelines, showing consistent improvements on both qualitative and quantitative benchmarks.

**Strengths:**

1. Simple, Effective, and General Idea: The core concept of using the difference between a weak and a strong model to approximate the direction of improvement is both intuitive and powerful. The framework's ability to operate on various types of "gaps" (weight, condition, etc.) demonstrates its impressive versatility.

2. Strong and Comprehensive Empirical Results: The paper provides extensive evidence of W2SD's effectiveness across multiple models (SD1.5, SDXL, DiT), tasks (image, video), and metrics (human preference, prompt adherence). The cumulative effect shown in Table 7 is particularly compelling.

3. Excellent Clarity and Presentation: The authors have done an outstanding job of explaining a complex idea in a clear and accessible manner. The writing and visualizations significantly aid in understanding the method's mechanics and motivation.

**Weaknesses:**

1. Idealized Assumption of the "Weak-to-Strong" Gap: The framework's core assumption is that the weak-to-strong gap vector is a reliable proxy for the strong-to-ideal gap. While this holds in the controlled 1D/2D experiments where models differ mainly in data bias, it may be too idealized for real-world scenarios. In practice, a strong and weak model may have qualitatively different failure modes or "worldviews" due to differences in architecture or fine-tuning data. In such cases, their difference vector could be noisy or even misaligned with the true direction of improvement. The paper could benefit from a discussion of this limitation.

2. Necessity of the Reflection Mechanism: The proposed reflective operation (denoise-then-invert) is elegant, but it is not immediately obvious if this complexity is necessary. A simpler alternative, such as directly guiding the strong model's prediction with the difference vector, might achieve similar results with less computational overhead. The lack of comparison against such a baseline makes it harder to isolate the contribution of the specific reflection mechanism.

**Questions:**

1. Regarding the mechanism itself: Could a simpler "difference guidance" approach work as well? For example, at each step, one could compute the predicted noise from both models, $\varepsilon_s$ and $\varepsilon_w$, and form a new prediction $\varepsilon_{\text{final}} = \varepsilon_s + \alpha \cdot (\varepsilon_s - \varepsilon_w)$, where $\alpha$ is a guidance scale. This seems to capture the same core idea but avoids the model inversion step. How does the proposed reflection operator compare to such a baseline?

2. Regarding the visualizations in Figures 3 & 4: Could you provide more specific details on how these plots were generated? For instance, how were the probability densities for the "Ideal," "Strong," and "Weak" models in Figure 3 obtained? For Figure 4, what were the specific data distribution ratios used to train the models to induce the bias? This would improve the reproducibility and clarity of this excellent intuitive demonstration.

3. Regarding the definition of "weak-strong" pairs: The paper shows many successful pairings. From your experience, are there examples of model pairs that are difficult to define as strictly "weak" or "strong," or pairs that failed to produce improvements? For example, what if one model is better at global composition but worse at fine details, while the other is the opposite? A discussion on such challenging or failure cases would provide a more complete picture of the framework's boundaries.

---

> ### Author Response · Authors · 2025-11-23
> **Response to Reviewer DEkr - part 1**
>
> We sincerely thank the reviewer for their constructive feedback and kind recognition of our work. Below, we provide detailed explanations to address your concerns.
>
> ---
>
> **W1**. Discuss limitations and case studies of the weak-to-strong gap as a proxy for the strong-to-ideal gap.
>
> We agree with the reviewer that further analysis would help clarify the underlying mechanism of W2SD.
>
> In our revised manuscript, **Theorem 2** rigorously establishes the relationship between the weak-to-strong gap (Δ₁) and the strong-to-ideal gap (Δ₂), including explicit error bounds for using Δ₁ as a proxy for Δ₂. **Fig. 3** visually demonstrates scenarios where this proxy relationship breaks down—specifically when Δ₁ and Δ₂ are directionally misaligned or exhibit significant magnitude mismatch.
>
> Furthermore, in comment Q3, we extend this analysis to real-world image generation—systematically identifying two failure cases where W2SD is not applicable, thereby clarifying the boundaries of weak-to-strong generalization in diffusion models.
>
> ---
>
> **W2**. Necessity of the Reflection Mechanism (e.g., simply using the interpolation of difference vectors as a substitute)
>
> We thank the reviewer for raising this point, which we have addressed in detail in **Appendix E.5** (Relationship with Auto-guidance). In fact, the iterative architecture of W2S offers several distinct advantages:
>
> - Over-saturation Mitigation：W2S can effectively mitigate over-saturation—an inherent limitation of interpolation-based methods **[1]**. As demonstrated in **Tab.16** and **Fig.30**, under the same settings (e.g., LoRA gap magnitudes), simple interpolation leads to severe over-saturation, resulting in performance across multiple metrics that is even worse than that of the base model.
>
> - Architectural Flexibility: Our method supports model gaps across different architectures (e.g., with varying latent shapes). We elaborate the results in **Fig.31**.
>
> - Iterative Refinement: The iterative nature of W2S enables multiple reflection operations at the same noise level (as implemented in our Supplementary Material source code), which is not feasible with interpolation-based methods.
>
> These advantages collectively confirm the value of the weak-to-strong iterative framework as a generalized and effective approach for model enhancement.
>
> ---
>
> **Q1**. Necessity of the Reflection Mechanism ?
>
> The reflection mechanism is indeed essential for achieving effective weak-to-strong guidance, as we elaborate in our response to comment W2.
>
> ---
>
> **Q2**. More details for Fig.3 and Fig.4 ? (In the revised version, please refer to Fig.4 and Fig.5)
>
> We appreciate the opportunity to clarify these experimental details. In Fig. 3, we present a motivational experiment using a bimodal 1D Gaussian mixture distribution as training data for our diffusion models. The specific configurations are as follows:
>
> - The strong model is trained on a GMM with mixture weights [0.6, 0.4] for components N(-4,1) and N(4,1), respectively.
>
> - The weak model is trained on a GMM with weights [0.7, 0.3] for the same components.
>
> - The ideal model uses the analytical solution of an equally-weighted GMM with weights [0.5, 0.5] to ensure theoretical precision.
>
> Fig. 4 follows the same experimental setup as Fig. 3. Additional visualization cases—including both successful and failed examples—are provided in **Fig. 25**.
>
> ---

---

> > ### Author Response · Authors · 2025-11-23
> > **Response to Reviewer DEkr - part 2**
> >
> > ---
> >
> > **Q3**. Are there pairs that failed to produce improvements?
> >
> > Yes, we acknowledge that such a scenario exists.  Following the reviewer's suggestion, we have conducted the following analysis. The newly added 1D visualization case in **Fig.3** demonstrates that when the weak-to-strong gap fails to meet the conditions specified in **Theorem 2**, it cannot serve as a reliable proxy.
> >
> > Furthermore, based on real-world data, we have empirically identified two scenarios where W2SD is not applicable:
> >
> > - Model conflict (see **Fig.22**): Occurs when the weak and strong models possess conflicting or mutually exclusive capabilities, leading to contradictory guidance signals that hinder effective optimization.
> >
> > - Model over-similarity (see **Fig.23**): Occurs when the capability gap between models is too small, resulting in an insufficiently weak guidance signal that fails to provide meaningful performance improvements.
> >
> > we concur with the reviewer that rigorously analyzing W2SD's performance in extreme scenarios will benefit future methodological development.  It's worth noting that in our main experiments, all evaluated model pairs maintain natural weak and strong distinctions (e.g., guidance/pipeline gaps). These configurations are both intuitive to identify and inherently resilient to such failure scenarios.
> >
> > ---
> >
> > We hope that these additional experiments and explanations resolve the reviewer's concerns and questions. We are more than happy to discuss if the reviewer has further questions or comments.
> >
> > ---
> >
> > [1] Sadat, Seyedmorteza, Otmar Hilliges, and Romann M. Weber. "Eliminating oversaturation and artifacts of high guidance scales in diffusion models." ICML 2025

---

### Official Review · Reviewer_RUsL · 2025-11-06

**Soundness:** 2
**Presentation:** 2
**Contribution:** 2
**Rating:** 4
**Confidence:** 3

**Summary:**

The paper introduces Weak-to-Strong Diffusion (W2SD), an inference-time method that alternates between denoising with a strong model and inverting (i.e. adding noise back) with a weak model. Using the weak-to-strong gradient gap ($\Delta_1 = \nabla \log p_s  - \nabla \log p_w$) to approximate the strong-to-ideal gap ($\Delta_2 = \nabla \log p_{\text{data}}  - \nabla \log p_s$) the authors demonstrate a training-free to improve sampling using a variety of different weak/strong model pairs.

**Strengths:**

**(S1)**: The approach is simple and training-free, and can flexibly be used with various different models.

**(S2)**: Experimental results explore a variety of different diffusion models and architectures.

**(S3)**: A discussion on compute-aware sampling is provided, with considerations for sampling quality within a fixed wall-clock budget. This is relevant for broader applicability.

**Weaknesses:**

**(W1)**: The proof of Theorem 1 is weak. There is no solid justification given to "neglect the approximation error" (L727). The proof ignores Jacobian terms or any formulation for the approximation error. No conditions are provided for when $\Delta_1 \approx \Delta_2$. Omitting these conditions is a significant weakness. This makes the justification more heuristic than theoretical.

**(W2)**: The authors note that when the gap magnitude is negative (i.e. strong model weaker than the weak model), quality degrades (L440). There are no provided principled diagnostics or automatic safeguards to prevent harmful pairings or schedules. This leaves automatic model-pair selection an open practical issue of the method.

**(W3)**: Incomplete baselines. There are strong, training-free inference baselines are only discussed but not thoroughly compared in tables: e.g., Auto-Guidance, Z-Sampling variants, TFG, other schedule- or guidance-optimization methods. Many tables compare only “base vs W2SD". These omissions cloud discussions of relative merit.

**(W4)**: Computational cost. The results in Figure 13 rely on a setting where $T_{w2s}$ is halved and $\lambda$ is small. There are multiple experiments in the appendix, however, that state $\lambda = T-1$, which implies the reflection operation (that uses two model passes) is performed at nearly every step according to Algorithm 1, effectively doubling the inference cost. Ablations on the reflection window and schedule are limited.

**(W5)**: Hyperparameter tuning. Based on Figure 11, hyperparameter choices can significantly influence quality. If the gap is negative due to poor choices for LoRA scales, guidance scales, or expert selection, then performance may not be good. This might require extensive tuning, which takes away from the "training free" approach of the paper.

**(W5)**: Omission of comparisons with “Reflected Diffusion Models.” The empirical and conceptual separation from related work like reflected diffusion models could be made more explicit.

Overall, the weaknesses outweigh the strengths. The paper would benefit from stronger baselines and a clearer theoretical justification.

**Questions:**

**(Q1)**: The term "reflective operator" is not well-justified. The operation $x_t = M_{inv}^{w}({M}^{s}(x_{t},t))$ is a "denoise-then-invert" step that applies a correction. The paper doesn't explain what is being "reflected" or from what surface/manifold.

**(Q2)**: When does $\Delta_1 \approx \Delta_2$ hold? Under what theoretical settings or formalized assumptions?

**(Q3)**: Are there simple, training-free heuristics to pick the weak and strong models, or the reflection window?

**(Q4)**: Could the authors report actual wall-clock time on a fixed GPU and per-image variance across seeds for several $T$ and $\lambda$?

**(Q)**: There are many minor typos (e.g., "applues," "socre," "proivde"), and sometimes ambiguity around whether $s_{\theta}$ is a score or an $\epsilon$-predictor.

---

> ### Author Response · Authors · 2025-11-23
> **Response to Reviewer RUsL - part 1**
>
> We sincerely appreciate the time and effort the reviewer has dedicated to providing thorough and constructive feedback on our work. Below, we provide detailed explanations to address your concerns.
>
> ---
>
> **W1**(a). No solid justification is provided for neglecting the approximation error.
>
> We thank the reviewer for this insightful question. We have specifically investigated this point in **Appendix E.3**, titled "The Impact of Approximation Error in the Inversion Process." As demonstrated in **Tab.14**, performance notably degrades when errors beyond the inherent "model gap error" are amplified.
>
> Moreover, in W2SD we employ a step-by-step inversion operation, which has been demonstrated to effectively suppress computational errors arising from approximation methods [1,2].
>
> ---
>
> **W1**(b). No conditions are provided for when Δ₁ = Δ₂.
>
> We thank the reviewer for this valuable suggestion. In response, we have added a more comprehensive and quantitative analysis of the conditions under which the gap proxy remains applicable in the revised manuscript.
>
> - We establish an analytical relationship between Δ₁ and Δ₂ in **Theorem 2**, which provides explicit criteria for Δ₁ to serve as a valid proxy for Δ₂. Specifically, the theorem shows that given exact analytical knowledge of any two of the three models—ideal, strong, and weak—the third can be derived analytically.
>
> -  **Fig. 3** provides visual validation of the theoretical criteria established in Theorem 2 through analytical solutions of a synthetic data distribution. Building on this theoretical framework, we further identify and characterize two distinct failure modes in real-image applications where W2SD becomes ineffective: Model Conflict (**Fig. 22**) and Model Over-Similarity (**Fig. 23**).
>
> - Furthermore, we present **Theorem 3** in the revised manuscript to prove that under the approximation conditions of Theorem 2, W2SD achieves a strictly lower Fisher Divergence compared to the standalone strong model. The total error satisfies $\mathcal{J}(p^{\mathrm{gt}} \| p^{\mathrm{w2sd}}) \leq \epsilon^2 \mathcal{J}(p^{\mathrm{gt}} \| p^{\mathrm{s}})$
>
> We hope these discussions further substantiate the theoretical contributions of W2SD, and we truly appreciate the opportunity to enhance our work through this valuable feedback.
>
> ---
>
> **W2**.Risk of Harmful Model Pairs in W2SD
>
> We fully understand the reviewer's concern regarding W2SD's performance in extreme scenarios with suboptimal model-pair configurations. It should be emphasized that the model pairs selected for our main experiments embody naturally emergent capability hierarchies—such as guidance and pipeline gaps—which reflect fundamental asymmetries inherent in knowledge systems. These configurations not only demonstrate intuitive identifiability but also inherent resilience to the failure scenarios under discussion.
>
> Additionally, in comment Q3 we introduce an enhanced variant of W2SD that incorporates a training-free heuristic for automatic model pair selection. Given a specific optimization objective—such as alignment with human preferences—This approach dynamically switches between model pairs during the denoising process, achieving enhanced adaptability while maintaining optimal performance.
>
> ---
>
> **W3**.Incomplete baselines (e.g. Auto-Guidance, Scheduler, Z-Sampling variants and others)
>
> We thank the reviewer for raising the point regarding baseline comparisons. In Appendix E.5, we provide a detailed evaluation demonstrating that W2SD consistently outperforms both Auto-Guidance [3] and Re-Sampling [4]—two closely related works in this domain.
>
> We would like to respectfully emphasize that W2S is conceived as a general-purpose framework, as systematically demonstrated in our meta-experimental overview (**Tab.1**), where all experiments are organized according to carefully designed model-pair configurations.  In contrast, methods such as Z-Sampling and its variants operate within more constrained single-model paradigms, which makes direct performance comparisons inherently challenging. Rather than positioning W2S as a narrow competitor, we highlight its potential to extend and enhance existing approaches—for instance, by applying pipeline gap refinements to Z-Sampling variants, thereby broadening their applicability and effectiveness.
>
> We thank the reviewer for their valuable suggestion. In future work, we plan to include more experimental results and comparisons with related methods to further demonstrate the effectiveness and general applicability of W2SD.
>
> ---
>
> [1] Mokady, Ron, et al. "Null-text inversion for editing real images using guided diffusion models." CVPR 2023.
>
> [2] Bai, Lichen, et al. "Zigzag diffusion sampling: Diffusion models can self-improve via self-reflection." ICLR 2025.
>
> [3] Karras, Tero, et al. Guiding a diffusion model with a bad version of itself. Neruips 2024.
>
> [4] Lugmayr, Andreas, et al. "Repaint: Inpainting using denoising diffusion probabilistic models." CVPR. 2022.

---

> ### Author Response · Authors · 2025-11-23
> **Response to Reviewer RUsL - part 2**
>
> ---
>
> **W4**.Ablations on the reflection window and scheduler are limited.
>
> We understand the reviewer's concern regarding efficiency, which is indeed an important consideration. We have added **Fig.19** and **Fig.20** to provide an ablation analysis of these two components.
>
> -  For the reflection window analysis, we conducted experiments with few-step (8 steps on a distillation diffusion model, Fig.19) and multi-step (20 steps on a standard diffusion model, Fig.20) settings, plotting the relationship between the reflection window size and HPS v2 metric. As a temporal inference scaling method, W2SD's performance naturally improves with increasing λ. Notably, significant gains are observed even with minimal reflection windows (e.g., λ=1).
>
>   | Ref Window | 0 (weak) | 0 (strong) | 1 | 2 | 3 | 4 | 5 | 6 | 7 |
>   |:----------:|:--------:|:----------:|:-:|:-:|:-:|:-:|:-:|:-:|:-:|
>   | λ/T Ratio | 0/8 | 0/8 | 1/8 | 2/8 | 3/8 | 4/8 | 5/8 | 6/8 | 7/8 |
>   | HPS Score | 31.4062 | 32.4687 | 32.5313 | 32.8750 | 32.9376 | 32.9687 | 33.0007 | 33.0112 | 33.0313 |
>
>   | Ref Window | 0 (weak) | 0 (strong) | 2 | 5 | 10 | 15 | 19 |
>   |:----------:|:--------:|:----------:|:-:|:-:|:-:|:-:|:-:|
>   | λ/T Ratio | 0/20 | 0/20 | 2/20 | 5/20 | 10/20 | 15/20 | 19/20  |
>   | HPS Score | 27.4531 | 30.6562 | 30.9375 | 31.3281 | 31.8281 | 32.2187 | 32.3125 |
>
> - For the scheduler analysis, we employed DPM Solver++ in **Fig. 19** and DDIM in **Fig. 20**, with results consistently demonstrating the robustness of our method across different scheduling strategies.
>
> ---
>
> **W5**. Doesn't the need for hyperparameter tuning contradict the "training-free" claim?
>
> We acknowledge the reviewer's concern and would like to clarify that the term "training-free" in this paper specifically refers to the absence of fine-tuning model parameters through gradient backpropagation (which typically requires substantial memory and computational time).
>
> Regarding the selection of inference hyperparameters, W2SD offers two advantages:
>
> - **Generalizable Inference Hyperparameters**: Once optimized for a model pair, W2SD's hyperparameters generalize effectively across diverse prompts and denoising steps. For example, **Tab.10** and **Tab.11** show consistent performance on two distinct prompt datasets (Pick-a-Pic and DrawBench) using identical hyperparameter configurations. For example, a quick grid search on just 5–10 cases yields hyperparameters that generalize well across diverse scenarios.
>
> - **Low-Cost Hyperparameter Search**: The inference hyperparameters can be optimized via grid search at minimal cost, as the process requires no gradient backpropagation and converges rapidly on a small validation set.
>
> ---
>
> **W6**.Omission of comparisons with “Reflected Diffusion Models.
>
> We sincerely thank the reviewer for bringing reference [5] to our attention.
>
> While both [5] and our W2SD approach employ the concept of "reflection", they are fundamentally distinct in their methodological foundations: [5] primarily utilizes reflected stochastic differential equations to reduce numerical discretization errors, whereas W2SD leverages collaborative guidance between dual models to achieve controlled distribution shifting. We have included a comprehensive discussion of these methodological distinctions in the related work section of our revised manuscript, along with expanded comparative analyses with additional relevant works.
>
> ---
>
> **Q1**. The definition of reflective operator
>
> We adopt the term "reflective operator" to describe the latent correction mechanism through strong denoising and weak inversion, aligning with the operational framework in Z-Sampling (FootNote[1], L107).
>
> ---
>
> **Q2**. When does Delta_1 = Delta_2 holds?
>
> The equality condition Δ₁ = Δ₂ holds when the weak-to-strong gap aligns perfectly with the strong-to-ideal gap in both direction and magnitude. We provide analytical proofs deriving sufficient conditions for this equality, supported by illustrative examples, refer to comment W1(b) for more details.
>
> ---
>
> [5] Lou and Ermon. Reflected Diffusion Models. ICML 2023.
>
> ---

---

> ### Author Response · Authors · 2025-11-23
> **Response to Reviewer RUsL - part 3**
>
> ---
>
> Q3.Are there simple, training-free heuristics to pick the weak and strong models, or the reflection window?
>
> Yes. We propose a W2SD variant (**Algorithm 2**) that introduces a simple heuristic for dynamically switching model pairs during denoising.
>
> This approach leverages a user-defined reward model (e.g., HPS or PickScore) to automatically designate strong and weak models at each denoising step. By evaluating intermediate outputs at noise level is zero, it enables real-time assessment and dynamic model selection, effectively overcoming the limitations of fixed pairing strategies. We report results in **Table 13** and **Fig. 24**, validating our dynamic model selection method.
>
> ---
>
> **Q4**: Could the authors report actual wall-clock time on a fixed GPU and per-image variance across seeds for several T and \lambda?
>
> Yes. We provide wall-clock time measurements conducted on an NVIDIA GeForce RTX 4090 24GB, in our experiments, the measured time demonstrates near-linear growth with λ, despite minor variations due to machine condition fluctuations. All configurations in this study utilize the guidance gap as the implementation of W2SD.
>
>
> | Ref Window (λ/T) | 0/20 | 1/20 | 3/20 | 5/20 | 7/20 | 9/20 | 11/20 | 13/20 | 15/20 | 17/20 | 19/20 |
> |:--------------:|:-:|:-:|:-:|:-:|:-:|:-:|:--:|:--:|:--:|:--:|:--:|
> | **Time per Image (s)** | 2.70 | 2.95 | 3.45 | 3.93 | 4.15 | 4.91 | 5.40 | 5.89 | 6.37 | 6.86 | 7.34 |
>
> | Ref Window (λ/T) | 0/50 | 10/50 | 20/50 | 30/50 | 40/50 | 49/50 |
> |:--------------:|:-:|:-:|:-:|:-:|:-:|:-:|
> | **Time per Image (s)** | 6.7950 | 8.7306 | 11.1672 | 13.6073 | 16.0537 | 18.2630 |
>
> ---
>
> Q5: There are many minor typos
>
> A: Thank the reviewers for highlighting these points. We will address all the identified issues in the revised version of our paper.
>
> ---
>
> Finally, we sincerely thank the reviewer RUsL for its diligent work and insightful comments. We hope that these additional experiments and explanations address the reviewer’s concerns, and we respectfully invite you to reconsider our submission.

---

### Meta-Review · Area_Chair_Meca · 2025-12-30

**Summary:**

The initial reviews were mixed, with three positive and two negative evaluations. During the rebuttal, most of the concerns raised by the negative reviews were adequately addressed. While the work still lacks strong theoretical justification, I believe the proposed framework offers a flexible and general approach to improving existing diffusion models, and its practical impact warrants acceptance.

**Reviewer Concerns:**

Reviewer 99Dw mainly raised concerns regarding presentation issues, such as figures and mathematical notation. These points were adequately addressed in the rebuttal and appear straightforward to fix. The reviewer also requested additional experiments with more recent models (e.g., W2SD on FLUX), which the authors have since provided. Reviewer RUsL raised concerns about the choice of baselines and computational costs, and these issues were also addressed during the rebuttal.

**Reviewer Scores:**

I believe that if the discussion period had been of normal length, the two negative reviewers would likely have been convinced by the authors’ responses, or at least would not have remained strongly negative. Overall, I think the strengths of the paper outweigh its weaknesses.

---

### Decision · Program_Chairs · 2026-01-26

Accept (Poster)